# Synergy of dual-atom catalysts deviated from the scaling relationship for oxygen evolution reaction

Cong Fang[1,2], Jian Zhou[1,2,3], Lili Zhang[1,2,3], Wenchao Wan[4], Yuxiao Ding [3,5] ✉ & Xiaoyan Sun [1,2,3] ✉

Dual-atom catalysts, particularly those with heteronuclear active sites, have the potential to outperform the well-established single-atom catalysts for oxygen evolution reaction, but the underlying mechanistic understanding is still lacking. Herein, a large-scale density functional theory is employed to explore the feasibility of *O-*O coupling mechanism, which can circumvent the scaling relationship with improving the catalytic performance of N-doped graphene supported Fe-, Co-, Ni-, and Cu-containing heteronuclear dual-atom catalysts, namely, M′M@NC. Based on the constructed activity maps, a rationally designed descriptor can be obtained to predict homonuclear catalysts. Seven heteronuclear and four homonuclear dual-atom catalysts possess high activities that outperform the minimum theoretical overpotential. The chemical and structural origin in favor of *O-*O coupling mechanism thus leading to enhanced reaction activity have been revealed. This work not only provides additional insights into the fundamental understanding of reaction mechanisms, but also offers a guideline for the accelerated discovery of efficient catalysts.

Oxygen evolution reaction (OER) is a vital process for water splitting, fuel cells, and metal-air batteries[1–3]. Essentially, the OER process is a thermodynamically uphill reaction with multi-step proton-coupled electron transfer, and the sluggish kinetics requires high-performance catalysts to reduce the reaction overpotential for practical applications[4–6]. $IrO_2$ and $RuO_2$, the present state-of-the-art OER catalysts[7–9], exhibit high OER activity but must be replaced due to high cost and lack of stability in order to facilitate wider practical applications[10]. A grant challenge for electrocatalysis is the design of OER electrocatalysts with high activity and durable stability.

Recently, an increasing amount of research has focused on single-atom catalysts (SACs) that inherit the advantages of both homogeneous and heterogeneous catalysts[11,12], particularly in N-doped graphene-supported single transition metal atoms

(M–N–C)[13,14]. Extension has been made from single intermediate electrochemical reactions i.e. the hydrogen evolution reaction (HER)[15] to multi-intermediate electrochemical reactions, such as oxygen reduction reaction (ORR)[16–18], nitrate reduction reaction ($NO_3$RR)[19], oxygen evolution reaction (OER)[20], and $CO_2$ reduction reaction ($CO_2$RR)[21–26]. This can be attributed to the benefits of SACs, which include an unsaturated coordination configuration, maximal metal utilization, and relatively uniform active sites[27]. However, the individual active sites determine the fixed adsorption mode, which is tightly correlated with the adsorption energy of multi-step reaction intermediates bonding on a single center. Especially for specific multi-step reactions like the OER process, the presence of a linear scaling relationship makes it rather challenging to simultaneously optimize and tune the adsorption of each oxygenated intermediate

[1]Qingdao Institute of Bioenergy and Bioprocess Technology, Chinese Academy of Sciences, 266101 Qingdao, China. [2]Shandong Energy Institute, 266101 Qingdao, China. [3]University of Chinese Academy of Sciences, 100049 Beijing, China. [4]Max-Plank Institute for Chemical Energy Conversion, Mülheim an der Ruhr 45470, Germany. [5]Lanzhou Institute of Chemical Physics, Chinese Academy of Sciences, 730000 Lanzhou, China. ✉e-mail: yuxiaoding@licp.cas.cn; sunxy@qibebt.ac.cn

species at the single active site[28], which seriously limits the design flexibility and catalytic activity[29].

Dual-atom catalysts (DACs), not only possess the advantages of SACs but also allow the tuning and optimization of multi-intermediate reactions owing to their synergistic effect, that is, more flexible dual-metal active sites can modulate the adsorption energy of specific intermediate species[30], which is well suited for complex catalytic processes with multiple reaction steps and intermediates. Experimental investigations have revealed that DACs could provide higher activity performance compared to SACs. For instance, atomically dispersed Ni–Zn pairs obtained >90% CO Faraday efficiency across the broad potential window from −0.5 to −1.0 V (vs. RHE), and the synergistic effect enabled a $CO_2$ electron reduction performance superior to Ni/Zn single-metal sites[31]. Theoretically, graphene-based DACs (denoted as M'M@NC) were predicted to promote different reactions such as ORR[32–34], NRR[35–37], HER[38,39], and $CO_2$RR[40–42]. Regarding the OER process, several M'M@NCs with successful synthesis and application have been reported[43–45], for example, the Co–Ni pair exhibited high OER electrocatalytic activity and kinetics in alkaline conditions[46]. Moreover, P-doped FeNi@NC showed a low overpotential of 250 mV at a current density of 10 mA cm$^{-2}$ [47]. Additionally, both DFT calculations[48,49] and machine learning[50–52] have made progress to explore and predict the superior performance of DACs in OER. Nevertheless, the combination of $M_2$ dimers experimentally applied to OER as well as the comprehensive and in-depth investigation of synergistic effects between metal–metal is still limited. In particular, the underlying reaction mechanism at the dual-metal active sites towards the enhanced OER activity remains to be further explored.

Recent studies have conclusively demonstrated that the contribution of unconventional mechanisms for electrocatalysis on the SACs cannot be overlooked and that deviations in scaling relationships significantly affect electrochemical activity, for instance, the formation of dihydride in HER[53], two *OH species in ORR and OER[54,55], as well as superoxo and peroxo complexes in OER[56]. Furthermore, the *O–*O coupling mechanism (OCM), as one of the unconventional mechanisms which were proposed on SACs[55], has also been shown with a remarkably different activity at the dual-metal active site from that of the conventional adsorption evolution mechanism (AEM)[48,57]. This is analogous to the direct coupling of adjacent oxygenated intermediates in the lattice oxygen mechanism (LOM), bypassing the generation of *OOH[58,59]. Moreover, unlike three-dimensional structured oxides that only allow the formation of *O–*O bonds on the surface due to the low bulk flexibility, the two-dimensional structure of graphene support can provide sufficient structural flexibility for the *O–*O coupling. Thus, one can reasonably assume that OER catalysis via OCM could potentially bypass the universal scaling relationship, leading to a lower reaction barrier and accelerated reaction kinetics[57,60]. From another point of view, DACs with two adjacent atomic metal species could act as one of the most representative platforms for directly investigating the chemical and structural origin of OCM.

However, to the best of our knowledge, there is still a lack of systematic study on screening and designing DACs following the OCM mechanism. In view of the myriad possible combinations of heteronuclear dimers, as well as the currently many unexplored DACs in the experiment, it is imperative to establish a feasible strategy, by which we could elucidate the fundamental factors in favor of OCM, popularize the advantages of OCM developed from valuable previous studies[54,55], and further predict the promising candidates with much improved OER activity. To this end, by means of large-scale density functional theory (DFT) calculations with a computational hydrogen electrode model, we herein established a series of heteronuclear diatomic catalysts (M'M@NC, M' = Fe, Co, Ni, Cu, and M is ranging from Ti to Au) as well as homonuclear transition metal diatomic catalysts $M_2$@NC embedded on a two-dimensional N-doped graphene surface. We successfully screened out four homonuclear and seven

heteronuclear DACs exhibiting superior OER activity via OCM than the corresponding SACs and the benchmark of minimum theoretical overpotential. Particularly, the detailed analyses of the reaction paths and electronic features of those promising DACs enable us to propose a general strategy in favor of the *O–*O coupling mechanism for designing high-performance OER catalysts, which could guide further exploration of an even broader composition space of DACs for other related electrocatalytic reactions.

## Results

### Geometric structure and stability of heteronuclear DACs

Considering the synthesis strategies of FeM, CoM, NiM, and CuM have been well-established and extensively reported in recent experimental studies[61,62], here, we choose Fe, Co, Ni, and Cu as the candidates for M'. Meanwhile, the metal dopants of M range from 3d, 4d, and 5d transition metals from IVB to IB (except Tc) with a total of 82 heteronuclear DACs. As displayed in Fig. 1a, the two transition-metal atoms ($M_2$) in DACs joined and were both surrounded by three pyridine $sp^2$-N atoms. Previous theoretical investigations demonstrated that this is the most energetically favorable configuration for embedding $M_2$ dimers in N-doped graphene, and various advanced characterization techniques comprehensively confirmed the atomically dispersed $M_2$ on the N-doped graphene as well as the presence of atomic pairs[31]. Accordingly, reasonably constructed M'M@NC catalyst models with metal–metal bond lengths in the range of 2.09 Å (CoV) to 2.75 Å (CuZr) were calculated (Fig. 1b and Supplementary Table 2). The bond lengths ranking of partial $M_2$ dimers is CuM > NiM > CoM > FeM, which is associated with the atomic radii of Fe, Co, Ni, and Cu, while the remaining bond lengths of $M_2$ dimers are independent of the atomic radii. Moreover, the computed bond lengths between $M_2$ dimers are corroborated with recent experimental measurements, e.g., the bond lengths obtained from both experimental and DFT calculations for the Ni–Zn pair in NiZn@NC is 2.5 Å[31], and the experimentally observed value of Ni–Cu bond length is 2.40 Å, which is also in agreement with our calculated value of 2.409 Å[63]. Thus, the rationality of our theoretical approach and the reliability of the present DFT method can be validated. Evidently, the dual-metal active sites with appropriate bond lengths can be expected to be advantageous in promoting O–O coupling with a low energy barrier.

The Bader charge analyses were also used to verify the electronic structural features of M'M@NC. The results show (Fig. 1c and Supplementary Table 3) that a substantial amount of charge is transferred from the $M_2$ dimers to the N-doped graphene, and the charge density difference of the models represented by FeZr@NC and FePd@NC exhibit a redistribution of charge (Supplementary Fig. 1), which results in the formation of strong-polarized and hybridized metal-N bonds. In addition, the DACs of FeM, CoM, NiM, and CuM all show a significant decreasing trend of charge transfer from left to right in the same period, which is consistent with the tendency of electronegativity (Supplementary Table 4)[64], indicating that the increased number of unpaired electrons in metal atoms favors the charge transfer to the surrounding atoms. Moreover, the charges transferred to the coordinated N atoms from all 86 $M_2$ dimers are 1.02–2.21|e|, considerably larger than 0.80|e| reported in the literature[65], which enables the $M_2$ dimers to exhibit high oxidation states.

Then, we evaluated the thermodynamic and electrochemical stability of the $M_2$ dimers. The possibility of the uniform distribution was determined by calculating the binding energy (BE = $E_{M'M@NC} - E_{M'M} - E_{NC}$) and aggregation energy ($E_{agg}$ = BE − $E_{coh}$, where $E_{coh}$ is the cohesion energy of the metal atom in their crystal) of the heteronuclear $M_2$ dimers with N-doped graphene, as displayed in Fig. 1d and Supplementary Tables 5–9[37,41]. The results show that the majority of $M_2$ dimers possessed sufficiently negative BE, indicating strong M–N bonds formation between the dispersed $M_2$ dimers and the surrounding ligand environment, which consistent with Fig. 1c.

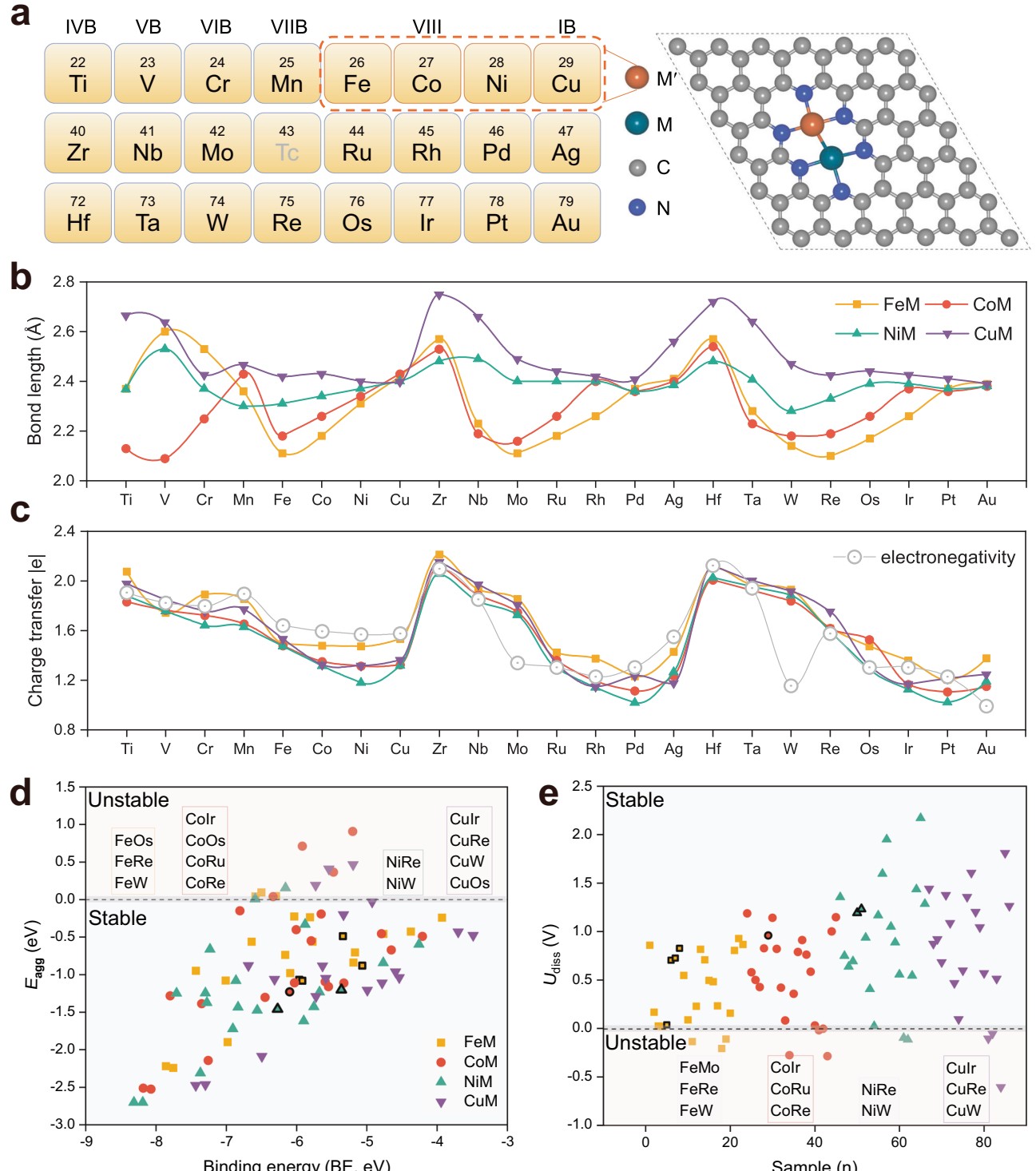

**Fig. 1 | Structural prototype, bond length, charge transfer, and stability of M′M@NC. a** Structural prototype of M′M@NC nanosheet. **b**, **c** Comparison of bond lengths and charge transfer of FeM, CoM, NiM, and CuM dimers on M′M@NC. **d** Computed binding energy (BE) and aggregation energy ($E_{agg}$) of metal dimers on N-doped graphene. **e** Calculated dissolution potential ($U_{diss}$) of M′M@NC at pH = 0, where the DACs systems highlighted in the black frame are the previously reported experimental ones (see discussion in the main text).

Furthermore, the calculated dissolution potential ($U_{diss}$) that applied to acidic solution show that most of the 86 $M_2$ dimers exhibited positive values in the range of 0–2.17 V (Fig. 1e and Supplementary Tables 10–13)[66], suggesting the stability of $M_2$ dimers in the acidic electrolyte and applied working potential environment under OER conditions. Overall, the $M_2$ dimers composed of FeM (Mo, W, Re and Os), CoM (Ru, Re, Os and Ir), NiM (Re and W), and CuM (W, Re, Os, and

Ir) need to be precautionarily averted in the experimental synthesis. Note that the DACs systems highlighted in black frame (Fig. 1d, e) are the reported experimentally synthesized metal dimers embedded in N-doped graphene, i.e., FeCo[67–69], FeNi[70–72], FeCu[73,74], CoNi[43,46], NiCu[63], Fe2[75], and Ni2[76,77], where BE, $E_{agg}$, and $U_{diss}$ fulfill our evaluation criteria as mentioned above, further demonstrating the reliability and feasibility of our approach.

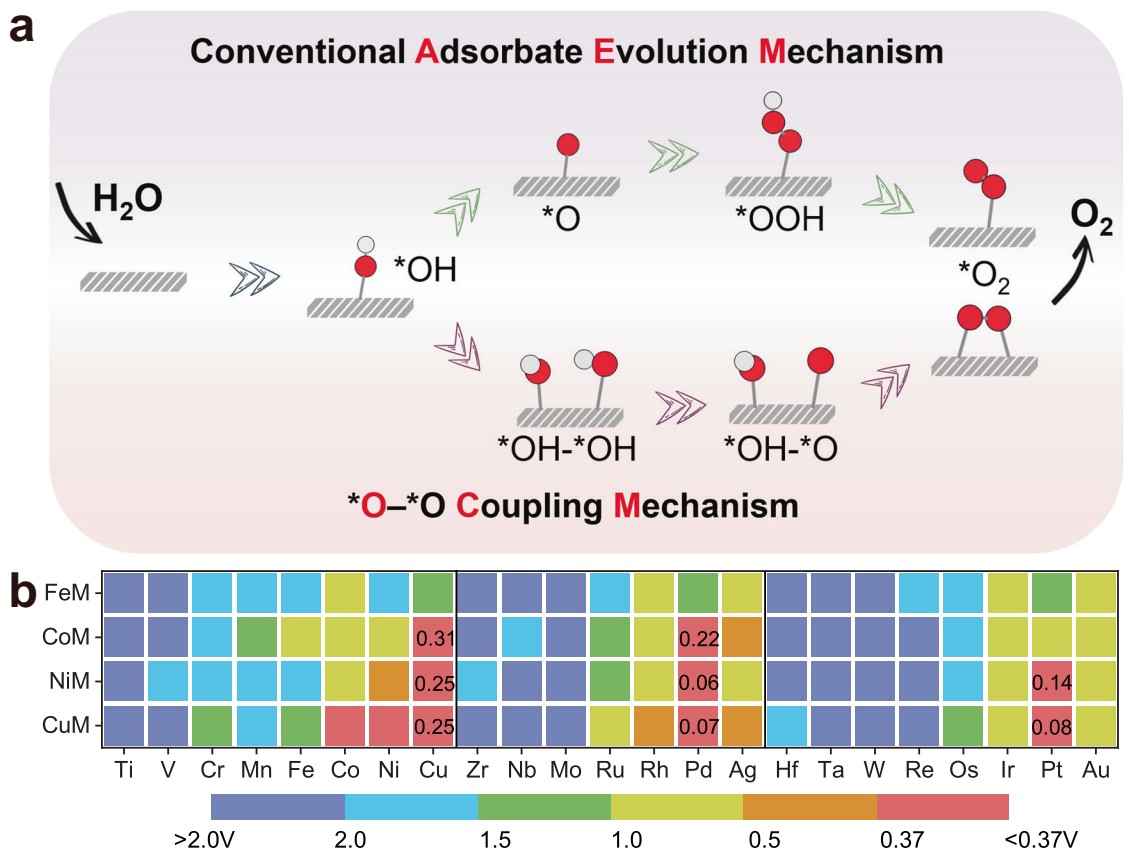

**Fig. 2 | Reaction mechanisms and activity map. a** Schematic illustration of the possible conventional adsorbate evolution mechanism (AEM) and O–O coupling mechanism (OCM). **b** The computed overpotential ($\eta$) of the FeM, CoM, NiM, and CuM dimers on M′M@NC.

## OER pathways and activities of DACs

In the conventional AEM that proceeds at a single active center, the OER involves multiple oxygen reaction intermediates, including *OH, *O, and *OOH, as displayed in Fig. 2a. The binding energies of these intermediates are highly correlated, leading to a minimum theoretical overpotential of about 0.37 V[60]. In contrast, OCM with the introduction of a second active center can break the structural constraint on the crucial oxygenated intermediate. In this case, the most significant difference (Fig. 2a) is that two adjacent adsorbed oxygen atoms have the possibility to generate one oxygen molecule by direct *O–*O coupling through the steps Eqs. (11)–(20). The sequence of two *OH adsorption, as well as deprotonation on the M2 dimer, followed the energetically favorable configuration and the eventual release of the coupled oxygen molecule into the next reaction cycle (Supplementary Tables 14–29). In principle, without the involvement of *OOH intermediate, OCM could circumvent the scaling relationship, and become the remarkably favorable reaction pathways with the advantages of dual metal active sites.

To establish a database of a series of crucial factors such as OER activity and adsorption of key intermediates, the potential energy surfaces of 86 DACs systems (FeM, CoM, NiM, and CuM) based on pre-selected OCM pathways were systematically explored. The bottleneck of the AEM mechanism ($\eta$ = 0.37 V) was used as a benchmark to manifest the high-performance OER catalysts and activity trends in the form of a heat map, as displayed in Fig. 2b and Supplementary Tables 30–60. Eight DACs meeting this benchmark were screened using $\eta$ < 0.37 V, and they exhibited appreciable catalytic activities toward the OER as NiPd (0.06 V) < CuPd (0.07 V) < CuPt (0.08 V) < NiPt (0.14 V) < CoPd (0.22 V) < NiCu (0.25 V) = CuCu (0.25 V) < CoCu (0.31 V), the computed total energies of oxygenated intermediates,

zero-point energy correction, and entropy contribution energy are shown in Supplementary Tables S31–S38. Interestingly, it is also clear in the heat map from left to right, the area of the red/blue area gradually increases/decreases, whereas the red area represents a lower $\eta$. Accordingly, the $M_2$ dimers composed of early-transition metals will hinder the progress of OER, either FeM@NC, CoM@NC, NiM@NC, and CuM@NC are not conducive to the performance. In contrast, $M_2$ dimers with the late-transition metals, especially these noble metals Pd, Pt, and Au, exhibit excellent OER activity and are well below 0.37 V via OCM reported so far. Note that CoNi@NC was experimentally reported with a low overpotential of 252 mV at 10 mA cm$^{-2}$, and not included in our screened systems[46], which is probably related to the different reaction mechanisms actually followed.

The eight $M_2$ dimers combinations with $\eta$ < 0.37 V were further compared to the corresponding M@NC (M = Co, Ni, Cu, Pd, and Pt) with similar N coordination environments to elucidate the activity differences between DACs and SACs, as well as the correlation between the synergistic effect of dual metal active site and structural engineering (Fig. 3 and Supplementary Figs. 2, 3). The computed total energy of oxygenated intermediates, zero-point energy correction, and entropy contribution energy of M@NC are given in Supplementary Tables 39–43. Based on the analysis of Gibbs free energy diagrams, potential determining steps (PDSs) of DACs following the OCM pathway can be found to primarily occur in the *O–*O coupling to generate *$O_2$ (Eqs. (14) or (19)), while the PDS of the six M@NC happens in the deprotonation of *OH to generate *O (Eqs. (2) or (7)) and the combination of *OH (Eqs. (1) or (6)). It is worth noting that the OER activities of all eight DACs are much higher than that of the corresponding SACs. Taking the best performing NiPd@NC as an example (Fig. 3d), the two *OH adsorbed on NiPd@NC with the most energetically favorable

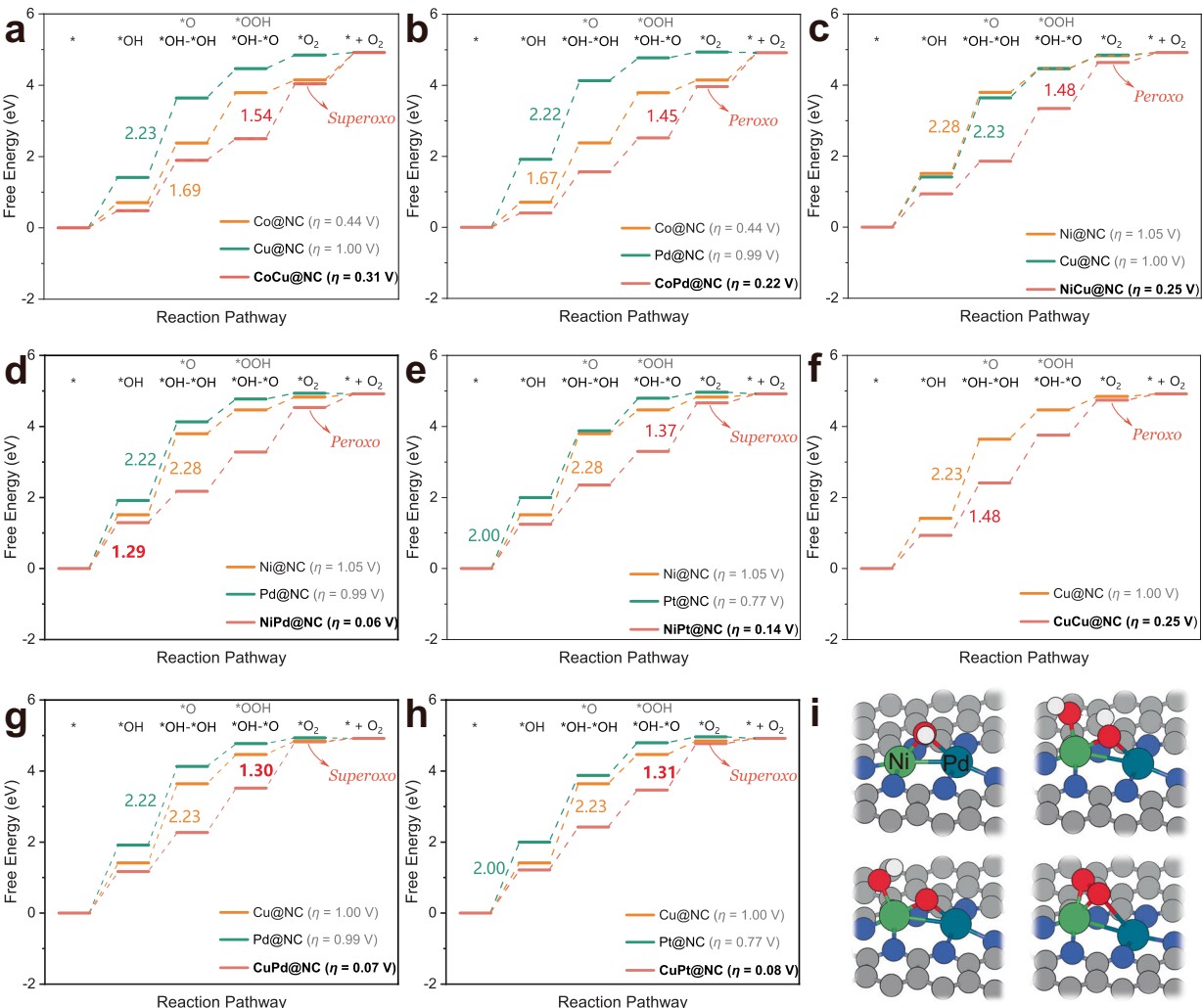

**Fig. 3 | Gibbs free energy diagrams and optimized configurations. a–h** Gibbs free energy diagrams of CoCu@NC, CoPd@NC, NiCu@NC, NiPd@NC, NiPt@NC, CuCu@NC, CuPd@NC, and CuPt@NC and the respective corresponding SACs for

OER. **i** Optimized configurations of oxygenated intermediates represented as NiPd@NC. The white, red, gray, and blue spheres in the atomic models represent H, O, C, and N atoms, respectively.

conformation; then *OH on the bridge site firstly fulfills the deprotonation to form *OH–*O in the bridging-$\mu^2$(M′, M) adsorption mode, which follows OCM with only slightly higher Gibbs free energy barrier than 1.23 eV to overcome the PDS (Eqs. (11) or (16)) and complete the reaction process. By contrast, Pd@NC and Ni@NC necessitate to overcome the Gibbs free energy barriers of 2.22 and 2.28 eV, respectively, to deprotonate *OH. To further verify the effectiveness of OCM, it is necessary to compare it with the conventional AEM for eight DACs (Supplementary Fig. 4 and Supplementary Tables 44–51). The PDS of NiPd@NC following AEM occurs at *OOH formation with a $\eta$ of 1.06 V, which is significantly higher than that via OCM. The remaining seven DACs systems exhibit similar phenomena as NiPd@NC. As a result, not only the effective catalysis of OER by dual metal active sites along OCM is more beneficial, but also the synergistic effect between $M_2$ dimers significantly improves the reaction activity compared to SACs.

The above analysis of activity is based on the hypothesis that both *OH are adsorbed on the metal dimer, and whether alternative adsorption behavior is possible? For example, the second *OH adsorbs on the adjacent C site, or on the reverse side of the catalyst[48,55]. As shown in Supplementary Fig. 5a and Supplementary Tables 52–55, virtually the entire DACs, the second *OH prefers to adsorb at the M–M site compared to the M–C site, which circumvents the effect of the adjacent C site for *OH adsorption. Likewise, we also investigated the

possibility of the second *OH adsorption on the other side of the DACs, which can be achieved for nearly half of the 86 DACs (Supplementary Fig. 5b and Supplementary Tables 52–55). However, although the above calculations indicate the possibility of *OH adsorption on the reverse side, we used a monolayer model for DACs in this work, whereas experimentally synthesized graphene-based catalysts usually consist of multilayers[78], and not only intercalation effects need to be considered, such structures also prevent *O–*O coupling[55]. On the other hand, different combinations of metal dimers have various O affinities without assurance that the *OH on the other side will not undergo further evolution (e.g., *O and *OOH), which complicates the picture and is not considered in this work.

It is worth noting that *O₂ → O₂(g), as a non-electrochemical process[79], is not appropriate to be included in the elementary steps, which overestimates the overpotential and misjudges the PDS. Inspired by the literature[56], *O₂ exhibits peroxide or superoxide characteristics in two different adsorption modes, end-on and side-on, respectively, due to the different combinations that induce variation in O-affinity. Supplementary Fig. 2 illustrates the specific adsorption configurations which are indicated in Fig. 3 as the peroxo- or superoxide nature of *O₂. On the other hand, the desorption of *O₂ was also evaluated for 86 DACs with a benchmark value of an experimentally reported DACs that follows OCM ($\Delta G_{*O2 \to O2(g)} = 1.17$ eV)[57]. The

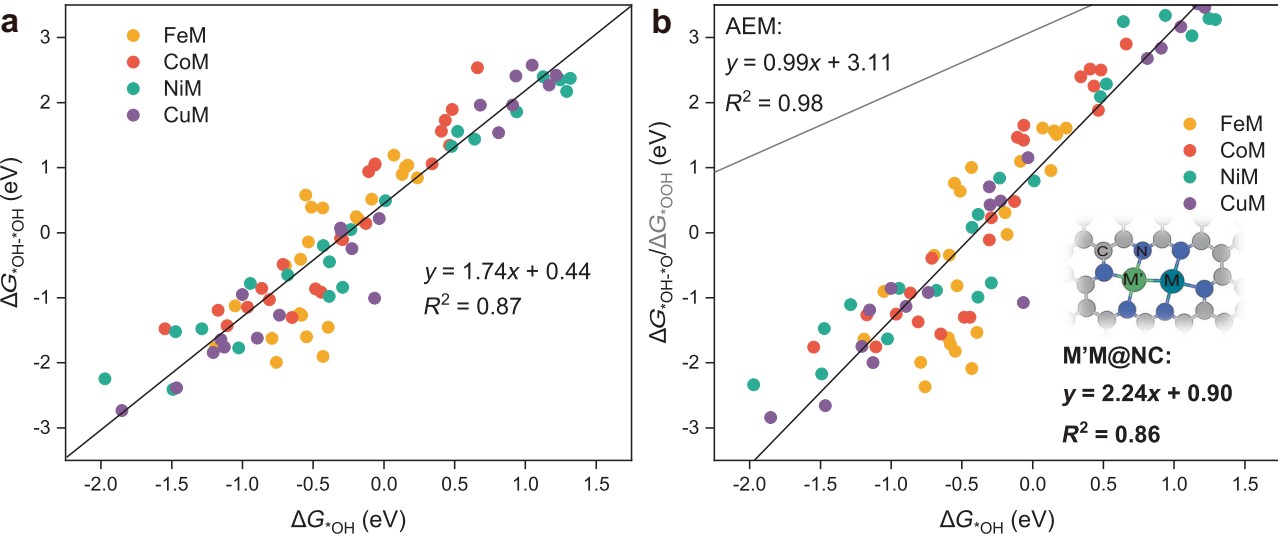

**Fig. 4 | Scaling relationships following the OCM. a** Relationship between $\Delta G_{*OH}$ and $\Delta G_{*OH-*OH}$ of M'M@NC. **b** Relationship between $\Delta G_{*OH}$ and $\Delta G_{*OH-*O}$ of M'M@NC. The gray line represents the scaling relationship following the AEM (containing SACs, DACs, and metal oxides, as shown in Supplementary Fig. 11). Inset shows the schematic configuration of M'M@NC.

excessive $\Delta G_{*O2\rightarrow O2(g)}$, i.e. higher than 1.17 eV, could be unfavorable for the reaction cycle, and it turns out that 8 high-performance DACs screened in the present study possess favorable desorption capacity with moderate O-affinity (Supplementary Table 56).

To better understand the significance of $*O_2$, a correlation between the charge amount of $*O_2$ received from the slab and the O–O bond length of $*O_2$ (Supplementary Fig. 6a) can be established, implying that the more charge $*O_2$ receives, the longer the bond length. Particularly, after excluding the disturbance data of DACs with $\Delta G_{*O2\rightarrow O2(g)} > 1.17$ eV, this linear correlation is more satisfactory (Supplementary Fig. 6b). Therefore, the O–O distance of $*O_2$ can be employed as a useful descriptor to describe the charge transfer. In addition, we also plotted the correlation with bond length and $\Delta G_{*O2\rightarrow O2(g)}$. Unfortunately, no strong correlation exists between the both, which is possibly caused by the wide energy range of the $\Delta G_{*O2\rightarrow O2(g)}$ (Supplementary Fig. 6c, d).

As the support also plays a vital role in electrochemical reactions[80], it is necessary to explore the activity of different support for the same metal dimers (FeM, CoM, NiM, and CuM). Two adjacent single vacancies on graphene accommodating metal dimers (M'M@Gr) were used as the prototype[81,82] (Supplementary Fig. 7) to further evaluate the support effect for the catalytic performance of DACs along the OCM. The activity trends of 86 M'M@Gr along the OCM for OER are shown as a heat map in Supplementary Fig. 8, with all oxygenated intermediates as the most stable configuration (Supplementary Tables 57–60). In analogy to Fig. 2b, the heat map also clearly shows a progressively improved activity from left to right, indicating that early-transition metal doping disadvantages the OER performance in either N-doped or defective graphene. In contrast to Fig. 2b, the quantities of DACs with high activity on defective graphene support are much less than that on N-doped graphene support, following the order FeFe(0.38 V) = CuPd(0.38 V) < CoAg(0.39 V) < NiCu(0.40 V) < CoPd(0.42 V) < CoCu(0.43 V) = CuCu(0.43 V) < NiAu(0.50 V). Based on the analysis of the free energy diagrams (Supplementary Fig. 9), the above 8 M'M@Gr exhibited strong adsorption of oxygenated intermediates, with consequently high overpotential and difficulty in $O_2$ desorption, thus unfavorable for the sustainability of reaction.

## Exploration of homonuclear DACs for OER

It is well known that in the conventional AEM, the difference between the adsorption-free energy of $*OOH$ and $*OH$ ($\Delta G_{*OOH}-\Delta G_{*OH}$) for an ideal OER catalyst is 2.46 eV, enabling all four elementary steps in the same magnitude of reaction free energy. However, the virtually constant difference ($\Delta G_{*OOH} - \Delta G_{*OH}$) of 3.2 ± 0.2 eV on the single active site has determined the lowest theoretical overpotential of 0.37 V ([3.2−2.46 eV]/2)[60]. Based on the above large-scale screening along the OCM, eight highly active DACs have overcome the limitation of minimum theoretical overpotential of 0.37 V. Obviously, the scaling relationship of OCM should be distinguished from that of conventional AEM. Consequently, we plotted the correlation between the adsorption-free energies of $*OH$, $*OH-*OH$, $*OH-*O$, and $*O_2$. Interestingly, there is a significant linear correlation between the adsorption-free energy of $*OH$ ($\Delta G_{*OH}$) and other oxygenated intermediates (Fig. 4 and Supplementary Fig. 10). Meanwhile, the dual metal active site followed the scaling relationship significantly deviated from that of the notably challenging AEM followed in Fig. 4b, without the constant difference of 3.2 ± 0.2 eV, which implicated that the minimum theoretical overpotential of 0.37 V was remarkably lowered by the dual metal active site, especially NiPd (0.06 V), CuPd (0.07 V) and CuPt (0.08 V). Notably, despite the fact that OER activity on M'M@Gr was inferior due to the excessively strong adsorption of oxygenated intermediates, the critical intermediates follow the same scaling relationship in Fig. 4b, albeit with different support (Supplementary Fig. 12), indicating a general rule of OCM on DACs.

The well-accepted Sabatier principle emphasizes that the binding of the adsorbed intermediates with the catalyst active site should be neither too strong nor too weak, both of which would impede the catalytic activity. $\Delta G_{*O} - \Delta G_{*OH}$ is generally employed as a descriptor for determining the OER activity in the volcano plot of AEM. Accordingly, we employed the similar descriptor $\Delta G_{*OH-*OH}$ for OCM to describe the catalytic behavior of OER on $M_2$@NC. Selection of descriptor is based on the fact that PDS occurs mainly in $*O-*O$ coupling that the adsorption of two $*OH$ determines the subsequent deprotonation as well as the $*O-*O$ coupling process. Interestingly, we found a volcano-shaped relationship between $\Delta G_{*OH-*OH}$ and $\eta$ (Fig. 5a), in which the optimal overpotential occurs as 1.56 eV $< \Delta G_{*OH-*OH} < 2.46$ eV, derived from eight DACs with $\eta < 0.37$ V (NiPd, CuPd, CuPt, NiPt, CoPd, NiCu, CuCu and CoCu). According to the volcano-shaped relationship, a strong combination of $*OH-*OH$ species in the left branch caused the $*O-*O$ coupling hardly achieved, resulting in the $O_2$ production as PDS. Conversely, the adsorption of $*OH$ or $*OH-*OH$ as PDS in the right branch implies that too weak

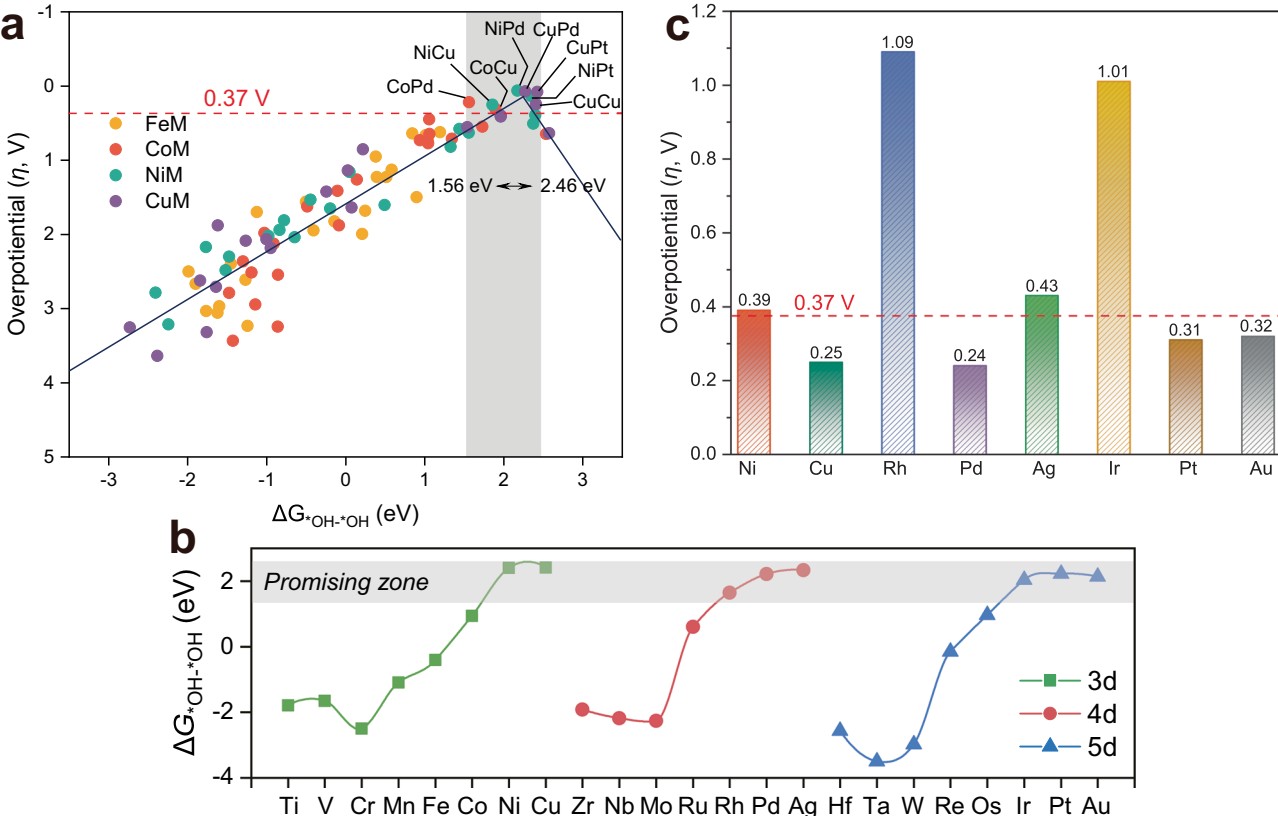

**Fig. 5 | Volcano-shaped relationship and predicted reaction activity of homonuclear DACs. a** Volcano-shaped relationship between the adsorption-free energy of oxygenated intermediates ($\Delta G_{*OH-*OH}$) and the theoretical overpotential ($\eta$). **b** Variations of $\Delta G_{*OH-*OH}$ on 23 homonuclear M$_2$@NC. **c** The $\eta$ of eight promising M$_2$@NC homonuclear DACs are summarized in terms of $\Delta G_{*OH-*OH}$ as the activity descriptor.

binding of *OH−*OH species with the active site prevents the adsorption of *O or *OH−*OH. Consequently, to balance both aspects the optimization of the $\Delta G_{*OH-*OH}$ through adjustment of the electronic structure enables to acquire of superior active catalysts.

Using the above $\Delta G_{*OH-*OH}$ as the activity descriptor, we extended our investigation to 23 homonuclear M$_2$@NC to validate this screening criterion. Figure 5b shows that eight of the 23 homonuclear M$_2$@NC were recognized as promising candidates, which were further analyzed with more details of catalytic activity. Ultimately, DFT calculations in Fig. 5c and Supplementary Fig. 13 confirmed four homonuclear M$_2$@NC with $\eta$ below 0.37 V: 0.24 V (Pd$_2$@NC) < 0.25 V (Cu$_2$@NC) < 0.31 V (Pt$_2$@NC) < 0.32 V (Au$_2$@NC), and the stability, optimized structures and the computed thermodynamic properties of oxygenated intermediates are given in Supplementary Fig. 14 and Supplementary Tables 61−69, respectively. Nevertheless, among the four excluded M$_2$@NC, Ni$_2$@NC, and Ag$_2$@NC approximated very closely to the criterion of 0.37 V as well, which illustrates the validity of the screening range of OCM from 1.56 to 2.46 eV. More importantly, this established descriptor could be used to accelerate developing more efficient catalysts, remarkably saving computational resources.

## Conditions for OCM and OER activity origin

All of the above calculations demonstrate that OCM is ideal for designing high-performance OER electrocatalysts. Here, critical questions are raised: is OCM suitable for all dual metal active sites? What is the condition that triggers the occurrence of OCM? Whether the crossings of two different mechanisms happen? To answer these questions, 105 DACs, including homonuclear composition, were explored in detail. The competition between second *OH adsorption and *OH deprotonation distinguished the two mechanisms, which was

first assessed by comparing Gibbs free energy change. Figure 6a and Supplementary Tables 70−74 show that 6 heteronuclear and 4 homonuclear highly active DACs screened prefer second *OH adsorption (except for CoCu) over competitive *OH deprotonation, suggesting that these DACs are selective for *OH−*OH rather than *O. Subsequently, based on a recent study[55], DACs with *OH−*OH selectivity were further analyzed to evaluate the possibility of intermediate crossover (*OH−*OH → *OH−*O or *OH−*OH → *OOH). Figure 6b and Supplementary Tables 75−79 show that all DACs with *OH−*OH selectivity proceed along the OCM without *OOH generation. This is because the *OH−*O conformation is energetically more stable than *OOH on the metal dimer structure, which explains the scaling relationship of OCM deviating from AEM in Fig. 4b. Notably, the absence of synergistic effects in the active center of SACs caused a different scaling relationship between *OH and *OH−*O along the OCM, without overlapping with that of DACs (Supplementary Fig. 15), which resulted in a complicated OER reaction network of the SACs.

For the DACs with *O selectivity in Fig. 6a, the intricate case of crossing intermediates between OCM and AEM induces the reaction to follow a hybrid pathway (*OH → *O → *OH−*O → *O$_2$ → O$_2$(g)). For the examples of DACs with *O selectivity shown in Supplementary Fig. 16 (selected as $\Delta G$(*OH−*O) > 0, Supplementary Tables 70−73), the PDS of all DACs except CoAu is *OH−*O → *O$_2$, and hybrid mechanism for crossing intermediates are only effective for $\Delta G$(*OH → *O) and $\Delta G$(*O → *OH−*O), therefore with limited effect on the catalytic activity of DACs with *O selectivity. Remarkably, the PDS of CoAu didn't occur at *OH−*O → *O$_2$, which enabled the reaction to bypass the PDS of *OH → *OH−*OH ($\Delta G$ = 1.88 eV), reducing the PDS to 1.25 eV (Supplementary Fig. 16c). This indicates that the increase in OER reactivity is feasible for CoAu via the crossing between OCM and AEM

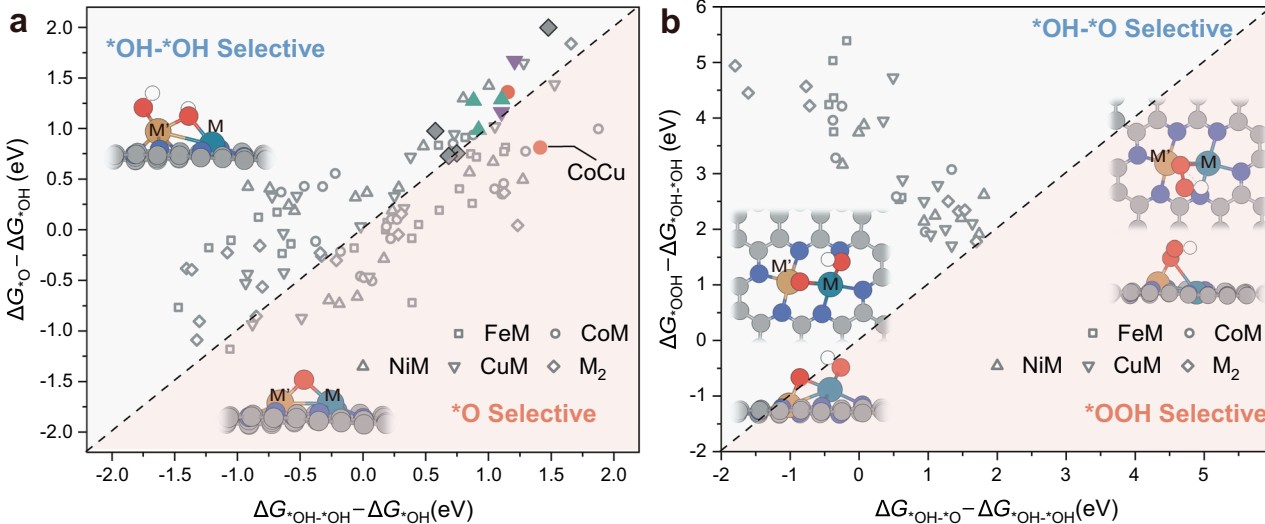

**Fig. 6 | Conditions for OCM. a** Gibbs free energy change ($\Delta G$) of *OH toward *OH–*OH versus *OH → *O + H$^+$ +e$^-$. Data points screened for high-performance homo- and hetero-nuclear catalysts are colored. **b** Gibbs free energy change ($\Delta G$) of *OH–*OH → *OH–*O + H$^+$ +e − versus *OH–*OH → *OOH + H$^+$ +e$^-$. Insets show the schematic configuration of the different intermediates adsorbed on M′M@NC. The white, red, gray, and blue spheres in the atomic models represent H, O, C, and N atoms, respectively.

intermediates. Additionally, the kinetic process of *O–*O coupling is also critical for triggering OCM, and higher kinetic barriers reduce the potential for OCM to occur. Therefore, we further calculated the possible coupling processes based on the 8 heteronuclear DACs screened (Supplementary Fig. 17 and Table 80). The results show that all 8 DACs possessed low coupling barriers and by the same magnitude as reported in the literature[48]. As a result, *O–*O coupling cannot hinder OCM from proceeding.

A critical question arises here: given that approximately half of the DACs exhibited *O selectivity, is it still valid to use $\Delta G_{*OH–*OH}$ in Fig. 5a as a descriptor to assess their reactivity? On the one hand, $\Delta G_{*OH–*OH}$ is generally applicable to OER processes along the OCM, and a secondary screening enables further rigorous identification of high-performance DACs with *O selectivity along hybrid mechanisms (e.g. CoCu). On the other hand, the fact that the $\Delta G_{*OH}/\Delta G_{*OH–*OH}$ scaling relationship and that of $\Delta G_{*OH}/\Delta G_{*O}$ virtually overlap rendered the DACs with $\Delta G_{*O}$ as descriptor still positioned on the original volcano curve (Supplementary Fig. 18). In other words, the volcano plot derived from $\Delta G_{*OH–*OH}$ with the screening scope from 1.56 to 2.46 eV still roughly determines the *O-selective DACs (e.g. CoAu) with regard to such accuracy, and the poorly active *O-selective DACs can be eliminated.

The adsorption of *OH not only plays a crucial role in determining the reaction mechanism but also the moderate adsorption strength can promote the same magnitude of free energy change for all the elementary steps, which approaches the ideal OER catalyst, rather than too strong/weak adsorption. Therefore, the DACs identified above with high OER activity should possess moderate interactions between intermediates and active sites. Here, we take the most active NiPd@NC as an example, the synergistic effect between metal dimers can be reflected in two aspects: (i) the modulation of the intrinsic electronic structure by introduction or substitution of additional metal atoms and (ii) the effect on the adsorption of the critical oxygenated intermediate *OH during the reaction.

First, we compared the density of states (DOS) of the most active NiPd@NC with the corresponding SACs (Ni@NC and Pd@NC). As shown in Supplementary Fig. 19a, the introduction of additional metal atoms seems to have a limited effect on the shape of the DOS, but the formation of M$_2$ dimers enables stronger adsorption of *OH than that of Ni@NC and Pd@NC, due to the charge redistribution bringing the DOS closer to the Fermi energy level ($E_f$) as shown in Supplementary

Fig. 19c, and causing the d-band center to shift upward. Also, for homonuclear combinations (Ni$_2$@NC or Pd$_2$@NC), the moderate modification of the electronic structure by the substitution of one atom leads to the alteration of the d-band center, e.g., the replacement of Pd in Pd$_2$@NC by Ni remarkably enhances the adsorption of *OH by the NiPd dimer from 1.47 to 1.29 eV (Supplementary Fig. 20).

The effect of the d-band center on the oxygenated intermediate *OH is reflected in the binding strength of metal−O bonds. Therefore, the bonding and antibonding states of *OH adsorption on the active centers of SACs and DACs were further analyzed by the crystal orbital Hamilton population (COHP) (Supplementary Fig. 19b). In contrast to Ni@NC and Pd@NC, the energy of the bonding state with NiPd@NC gradually shifted away from $E_f$ and the filling of the antibonding orbitals gradually decreased, which contributed to the increase of the *OH adsorption strength. To gain quantitative insight into the d−2p interaction, the integrated crystal orbital Hamiltonian population (ICOHP) analysis was calculated by integrating the states of the highest occupied energy levels. Interestingly, excellent linear correlations between $\Delta G_{*OH}$ values and ICOHP, as well as the d-band center, were observed (blue line in Supplementary Fig. 21), which shows that the closer d-band center to $E_f$ leads to the stronger *OH binding with more contribution from the occupied bonding states. This linear correlation quantitatively demonstrates that the synergistic interaction can readily adjust their electronic structures by d-band center and the adsorption nature of active sites, which are highly related to the intrinsic OER activity observed in Fig. 3d and Supplementary Fig. 20. The electronic structures of NiPt, CuPd, and CuPt also have similar phenomena and regularities, as shown in Supplementary Fig. 22.

In addition, we further extended to the remaining 10 high-performance DACs, where a similar linear correlation of $\Delta G_{*OH}$ and the d-band center was observed. It is worth noting that the d-band center for Cu-containing dimers (red line in Supplementary Fig. 21a) is much more negative and deviating severely from that of NiM dimers. This could be due to the neglected effect of the d-band shape and the sp orbitals of the metal sites[83,84], which further resulted in the opposite trend in the correlation of $\Delta G_{*OH}$ and ICOHP (red line in Supplementary Fig. 21b). Notwithstanding, it is evident that the synergistic effect of metal−metal sites render the possibility to regulate the electronic structure and d-band center. This consequently enables the binding intensity of *OH to the active site within a mild range

(−1.70 < ICOHP < −1.20 for high-performance $M_2$ dimers, except $Au_2$) and detrimental to deprotonation, thus eventually benefiting the Gibbs free energy barriers of all four elementary steps to be closer to the ideal 1.23 eV. We also plotted the M1−M2 distance versus $\eta$, as shown in Supplementary Fig. 23, with the appropriate range favoring the occurrence of OCM.

## pH-dependent and potential-dependent OER activity

The above results based on conventional constant charge methods (CCM) demonstrate that the synergistic effect between the dual-metal active sites in DACs can enhance OER activity. To better simulate the electrochemical interface and to examine the effect of pH as well as potential on the reaction activity, we took advantage of the JDFTx code of the grand canonical DFT (GC-DFT) for the constant potential method (CPM) calculations[85], which have been proved useful in recent electrochemical studies[86,87]. As an example, with the screened high-performance NiPd@NC, CuPd@NC, and CuPt@NC, we adjusted the absolute electrode potential to change the pH and electrochemical interface potential (referenced to RHE) as shown in Fig. 7 and

Supplementary Tables 81−83. Interestingly, the activity order evaluated by the two methods of CCM (Fig. 3) and CPM was consistent for the three DACs under alkaline conditions. In addition, the change in pH triggered a change in PDS, rendering the higher $\eta$ for CuPd than CuPt under acidic conditions. The order of $\eta$ for these three DACs is 0.08 V (NiPd) < 0.18 V (CuPt) < 0.20 V (CuPd) in acidic conditions, and 0.10 V (NiPd) < 0.18 V (CuPd) < 0.28 V (CuPt) in alkaline conditions, respectively, indicating they are all potentially promising catalysts via OCM under both acidic and alkaline conditions.

As shown in Fig. 7a–f, the free energy changes ($\Delta G$) of the elementary steps decreases as the applied potential increases until the overpotential is determined with the maximum $\Delta G = 0$ (Supplementary Fig. 24), and this is analogous to a recent study of $(phen_2N_2)FeCl$ monolayer[88]. Furthermore, the PDS of the three DACs is *OH–*O → *O$_2$ (for NiPd@NC and CuPd@NC at pH = 1, it is *OH–*OH → *OH–*O). The difference in activity of the three DACs derived from the different sensitivity of their oxygenated intermediates to the electrical potential, with the *OH–*O species being more significant (Supplementary Fig. 24). The decreasing trend of $\Delta G$ (*OH *O → *O$_2$) can be prominently

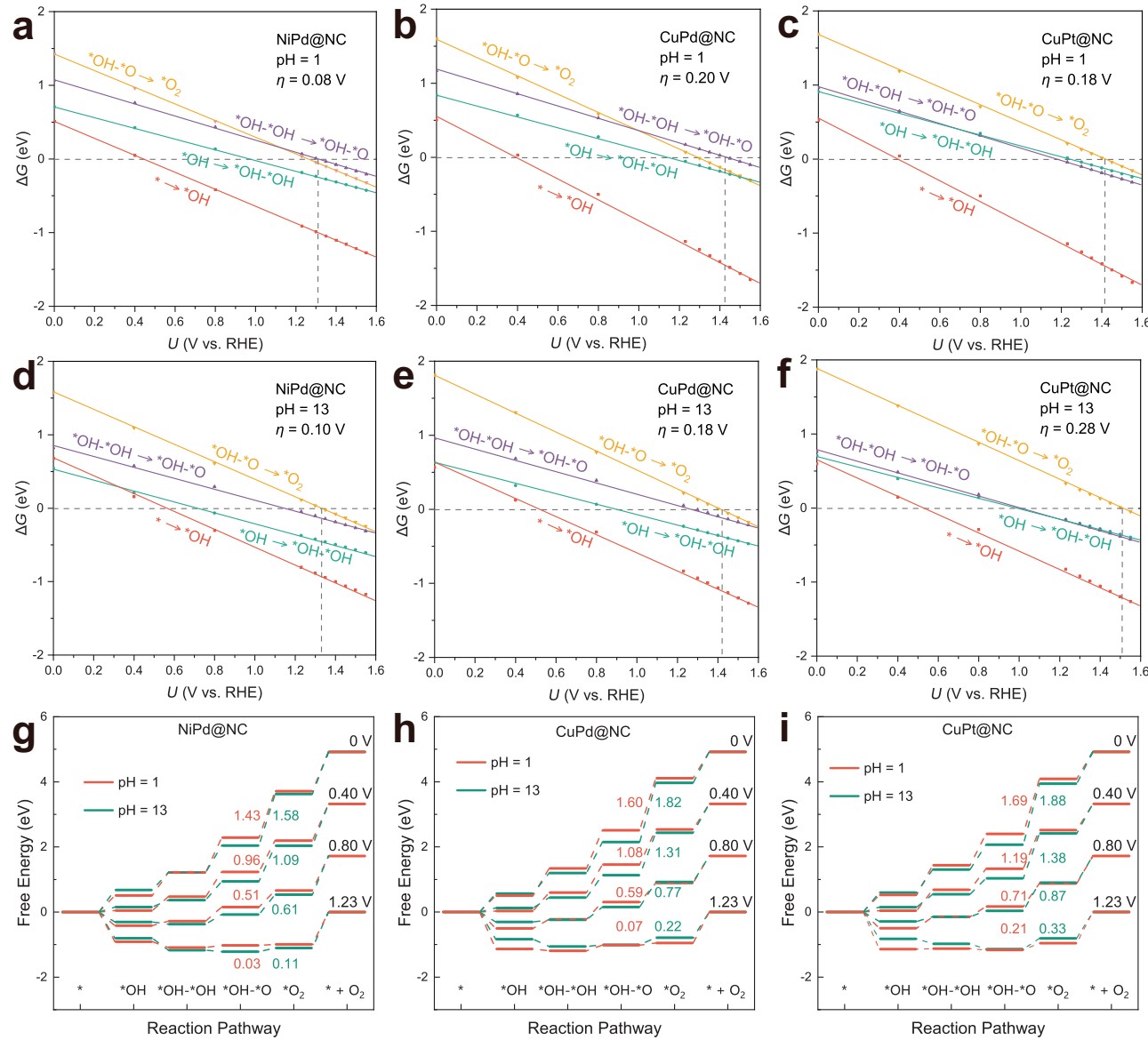

**Fig. 7 | Free energy diagrams for pH-dependent and potential-dependent.** Variation of Gibbs free energy with potential for **a** NiPd@NC, **b** CuPd@NC, and **c** CuPt@NC at pH = 1. Variation of Gibbs free energy with potential for **d** NiPd@NC,

**e** CuPd@NC, and **f** CuPt@NC at pH = 13. Free energy diagrams of **g** NiPd@NC, **h** CuPd@NC, and **i** CuPt@NC at different potentials with pH = 1 and 13, and the corresponding $\Delta G_{*OH\cdot*O\rightarrow*O2}$ are labeled.

observed in Fig. 7g–i with the gradual application of electrical potential. In terms of pH, another critical factor affecting activity, the considerable difference of 0.1 V in the overpotential of CuPt@NC under acidic and alkaline conditions can be attributed to the shift in the relative location of *OH–*O → *O$_2$ (as denoted by the yellow lines in Fig. 7c and f). The PDS of NiPd@NC and CuPd@NC altered from *OH–*O → *O$_2$ to *OH–*OH → *OH–*O as the alkaline to acidic conditions changed (Fig. 7a and b). Consequently, both pH and potential have significant contributions to the electrochemical process, with the influence on the PDS and overpotential. Nevertheless, the superior OER activity for the screened three DACs can be appraised.

## Discussion

In summary, we employed M'M@NC catalyst as a prototype to explore the activity and feasibility of OER along the OCM on the dual-metal active site. Large-scale screening constructed activity maps and predicted NiPd@NC (0.06 V), CuPd@NC (0.07 V), and CuPt@NC (0.08 V) significantly lower than the bottleneck 0.37 V of conventional AEM. The more favorable configuration of adsorption on the M$_2$ dimer for two *OH, which tactfully circumvents the scaling relationship between *OOH and *OH, renders $\eta$ closer to the ideal 0 V. Further, rationally constructed descriptor $\Delta G_{*OH-*OH}$ for the prediction of homonuclear compositions verified the effectiveness of the design strategy. More importantly, the feasible occurrence of OCM could be assessed based on the Gibbs free energy barriers of the two competing reactions (*OH → *OH–*OH vs. *OH → *O). We further demonstrated that the synergistic effect between M$_2$ dimers efficiently modulates the d-band center as well as the bonding/anti-bonding state of M–O bonds to facilitate the moderate adsorption of *OH, thus enabling high-performance DACs via OCM. This work provides a comprehensive understanding of the OCM mechanism, rendering an alternative recipe for the catalytic optimization of OER, with a volcano top different from that of conventional AEM. Further, CPM considering surface charge reveals pH-dependent and potential-dependent OER activity on diatomic catalysts, indicating the selected three DACs of NiPd@NC, CuPd@NC, and CuPt@NC are promising candidates under both acidic and alkaline conditions. It is important that OCM should be considered as a prominent OER pathway catalyzed by DACs in future studies. These theoretical insights would be also beneficial in designing high-performance catalysts for other electrochemical reactions.

## Methods

### DFT calculations

All spin-polarized density functional theory (DFT) calculations were performed using VASP 6.2.0 code[89]. The exchange-correlation function was employed to accurately simulate the interactions between electrons, which is described by the Perdew–Burke–Ernzerhof (PBE)[90] within the generalized gradient approximation (GGA-PBE). Taking 450 eV as the structural relaxation cutoff energy and the Gaussian smearing with $k_B$T = 0.05 eV to accelerate convergence, the core–valence interactions in the system were described in terms of the projector augmented wave (PAW) pseudopotential[91,92]. A series of Fe-, Co-, Ni-, and Cu-based heteronuclear and homonuclear metal dimers are embedded in N-doped graphene (NC), which were built based on a $p(6 \times 6)$ monolayer of pristine graphene, for which M@NC was constructed using a $p(5 \times 5)$ monolayer slab. A vacuum layer with a distance of 15 Å was employed in the vertical direction of the two-dimensional monolayer to avoid the interaction between two adjacent periodic images. The Γ-centered Monkhorst–Pack scheme[93] with 3×3×1 and 5×5×1 $k$-points grids was adopted to sample the Brillouin zone in the reciprocal space for geometric optimization and electronic structure calculations, respectively. The ionic optimization is performed using the conjugate gradient algorithm with the electron relaxation convergence threshold set to $10^{-5}$ eV and the Hellmann-Feynman force convergence threshold set to 0.01 eV/Å. In addition, the solvation effect

of water was simulated with the implicit Poisson–Boltzmann solvation model of VASPsol[94], and the long-range van der Waals interactions were performed using empirical dispersion (DFT-D2)[95]. The calculation of the charge transfer between atoms was adapted from the Bader charge analysis method[96]. The VASPKIT code was used to post-process the results of the VASP calculations[97]. The climbing image nudged elastic band (CI-NEB) method[98] combined with improved dimer method (IDM)[99] was employed to searched for transition states (TS).

### Free energy computations

The adsorption evolution mechanism (AEM) was widely accepted as a theoretical approach to mimic the OER reaction in acidic (pH = 1) and alkaline solutions (pH = 13)[100], which can be described as a four-electron transfer reaction pathway with the overall reaction pathway of OER in acidic and alkaline solutions as shown in Eqs. (1)–(5) and (6)–(10), respectively:

$$* + H_2O(l) \rightarrow *OH + H^+ + e^- \tag{1}$$

$$*OH \rightarrow *O + H^+ + e^- \tag{2}$$

$$*O + H_2O(l) \rightarrow *OOH + H^+ + e^- \tag{3}$$

$$*OOH \rightarrow *O_2 + H^+ + e^- \tag{4}$$

$$*O_2 \rightarrow O_2(g) + * \tag{5}$$

$$* + OH^- \rightarrow *OH + e^- \tag{6}$$

$$*OH + OH^- \rightarrow H_2O + *O + e^- \tag{7}$$

$$*O + OH^- \rightarrow *OOH + e^- \tag{8}$$

$$*OOH + OH^- \rightarrow *O_2 + e^- \tag{9}$$

$$*O_2 \rightarrow O_2(g) + * \tag{10}$$

where *, *OH, *O, *OOH, and *O$_2$ denote adsorbed sites, adsorbed oxygenated intermediate species OH, O, OOH, and O$_2$, respectively. In addition, the *O–*O coupling mechanism (OCM) on dual-metal active sites in acidic (Eqs. (11)–(15)) and alkaline solutions (Eqs. (16)–(20)) is a feasible reaction pathway[48]:

$$* + * + H_2O(l) \rightarrow * + *OH + H^+ + e^- \tag{11}$$

$$* + *OH + H_2O(l) \rightarrow *OH - *OH + H^+ + e^- \tag{12}$$

$$*OH - *OH \rightarrow *OH - *O + H^+ + e^- \tag{13}$$

$$*OH - *O \rightarrow *O_2 + H^+ + e^- \tag{14}$$

$$*O_2 \rightarrow * + * + O_2(g) \tag{15}$$

$$* + * + OH^- \rightarrow * + *OH + e^- \tag{16}$$

$$* + *OH + OH^- \rightarrow *OH - *OH + e^- \tag{17}$$

$$*OH - *OH + OH^- \rightarrow *OH - *O + H_2O + e^- \qquad (18)$$

$$*OH - *O + OH^- \rightarrow *O_2 + H_2O + e^- \qquad (19)$$

$$*O_2 \rightarrow * + * + O_2(g) \qquad (20)$$

where $*O_2$ includes the side-on configuration ($*O-O*$) as well as the end-on configuration ($*OO$), both of which employed the most stable configuration of the adsorbed $O_2$ in the calculations. The free energy diagrams of the OER elementary steps involving ($H^+ + e^-$) pair transfer as proposed by Nørskov and co-workers were calculated using the computational hydrogen electrode (CHE) model[101]. The Gibbs free energy ($\Delta G$) changes for each elementary step are given as follows:

$$\Delta G = \Delta E + \Delta E_{ZPE} - T\Delta S + \Delta G_U + \Delta E_{sol} \qquad (21)$$

where $\Delta E$ is the total energy of the reaction under vacuum conditions calculated by DFT, with $\Delta E_{ZPE}$ and $T\Delta S$ representing the zero-point energy change based on the calculated vibrational frequency and the entropy change at a reaction temperature at $T = 298.15$ K, respectively. The values of $E_{ZPE}$ and TS for $H_2O$ and $H_2$ can be seen in Supplementary Table 1. The water dielectric constant ($\varepsilon = 80$) was adopted to take into account the effect of the solvent on oxygenated intermediates and is denoted by $\Delta E_{sol}$. $\Delta G_U = -neU$ is the contribution of the applied electrode potential to the reaction Gibbs free energy, where $n$ and $U$ are the number of transferred ($H^+ + e^-$) pairs and the applied electrode potential versus the reversible hydrogen electrode (RHE) for each elementary reaction, respectively. In addition, the energy of the open-shell triplet ground state $O_2$ molecule is derived as $G_{O2} = 2G_{H2O} - 2G_{H2} + 4.92$ eV due to the inability of DFT to precisely calculate the energy of the $O_2$ molecule, while the entropy of the gaseous molecules ($H_2$ and $H_2O$) was taken from the NIST Chemistry WebBook[102]. For OCM the Gibbs free energy of the elementary steps is calculated as

$$\Delta G_1 = \Delta G_{*OH} - eU \qquad (22)$$

$$\Delta G_2 = \Delta G_{*OH-*OH} - \Delta G_{*OH} - eU \qquad (23)$$

$$\Delta G_3 = \Delta G_{*OH-*O} - \Delta G_{*OH-*OH} - eU \qquad (24)$$

$$\Delta G_4 = \Delta G_{*O2} - \Delta G_{*OH-*O} - eU \qquad (25)$$

$$\Delta G_5 = 4.92 \text{ eV} - \Delta G_{*O2} \qquad (26)$$

The theoretical overpotential was derived from the following equation:

$$\eta = \max[\Delta G_1, \Delta G_2, \Delta G_3, \Delta G_4]/e - 1.23 \text{ V} \qquad (27)$$

## Constant-potential calculations

The grand canonical ensemble DFT simulations were performed using the JDFTx software[85], and the electron exchange-correlation energy was evaluated within the GGA-PBE functional, consistent with the calculation in VASP. The GBRV pseudopotentials and a plane wave energy cut-off of 20 Hartree was employed. The linear PCM solvation model (CANDLE) simulates the liquid environment, with 0.1 M $K^+$ and $F^-$ ions[103]. The Brillouin zone was sampled using a $2 \times 2 \times 1$ Monkhorst–Pack $k$-point mesh and a convergence threshold of $1 \times 10^{-6}$

Hartree was set for the total electronic energy. The effective regulation of pH and potential can be described by the following relation:

$$U(V/RHE) = U(V/SHE) + 0.0592 \times pH \qquad (28)$$

$$U(V/SHE) = -4.66 \text{ V} - U_{applied}(V/SHE) \qquad (29)$$

where $-4.66$ V is the absolute electrode potential with SHE as reference[104], and the $U_{applied}$ is the applied electrode potentials of the targets.

## Data availability

The data that support the findings of this study are available from the corresponding author upon request. Source data are provided with this paper. The atomic coordinates of the optimized model for electronic structure calculations are provided as a separate Supplementary Data 1. Source data are provided with this paper.

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

## Acknowledgements
X.Y.S. acknowledges the financial support of Shandong Energy Institute Fund (Grant No. SEII202122), QIBEBT (Grant: QIBEBT S202306), Natural Science Foundation of Shandong Province (ZR2021QB128, ZR2022MA040), and Shandong Province Taishan Young Scholars Project (tsqn201909161). Y.X.D. acknowledges the financial support from the National Natural Science Foundation of China (22202217). We thank Dr. Shanjun Mao from Zhejiang University, Prof. Yafei Li, and Dr. Tianyang Liu from Nanjing Normal University for their helpful discussions. We also thank Prof. Wei An and Yunyi Li from the Shanghai University of Engineering Science for their assistance with the JDFTx software for constant potential method calculations. The numerical calculations in this paper have been done at Hefei's advanced computing center.

## Author contributions
X.Y.S. and C.F. conceived and directed the research. C.F. conducted the calculations and analyzed the data. W.C.W. and Y.X.D. discussed the calculation results and commented on the paper., J.Z. and L.L.Z. contributed to the discussions. C.F. prepared the manuscript with feedback from all authors.

## Competing interests
The authors declare no competing interests.
