## [Peer Review File · Nature Communications]

Synergy of Dual-atom Catalysts Deviated from the Scaling Relationship for Oxygen Evolution ReactionEditorial Note: Parts of this Peer Review File have been redacted as indicated to remove third-party material where no permission to publish could be obtained.

REVIEWER COMMENTS

Reviewer #1 (Remarks to the Author):

In this work the authors presented a computational study on the catalytic activity of dual atom catalysts for the electro-chemical oxygen evolution reaction, finding some recipes for the catalyst optimization, that could be of help for both the theoretical and experimental design of novel materials.

I must say that current level of investigation, and generality of the message is not sufficient for Nature Communications. If the author could provide further and broader evidence of the following major points, I will be happy to recommend publication of the work.

1) The take-home message is unclear, and this strongly harm the potential impact of the work. More specifically, it is unclear whether the main message is one or a combination between the following:

1.1 OER does not follows the typical mechanism.

1.2 The authors are providing a recipe for the catalytic optimization, with a volcano top different from that of conventional catalysts.

1.3 New scaling relationships are provided for OER on DACs.

Each of the mentioned points should be better corroborated by additional calculations or acknowledgement of the existing literature.

1.1 The message is not totally new, since it has been recently demonstrated that OER could follow the alternative path proposed by the authors although with just a single metal center (single-atom catalysts) [10.1016/j.jcat.2022.12.014; 10.1021/acscatal.2c03020; 10.1021/acscatal.0c00815]. The fact that SACs can form molecular complexes, as coordination chemistry compounds is a very important hallmark of this class of materials, and it is actually more general than OER [10.1002/smt.201800368; 10.1021/jacs.1c10470]. The mentioned literature should be acknowledged.

1.2 The authors should better corroborate the findings. More specifically, I am missing if the authors checked if the unconventional intermediates are more stable than the classical ones. This is an essential point, given that it is likely than the reaction will follow a reaction channel based on the most stable intermediates. Also, the picture can be more complex, with the crossing between conventional and unconventional intermediates [10.1016/j.jcat.2022.12.014]. This could affect the scaling relationships plots, since one should consider only the most stable intermediate for each electron transfer. This also means that the assumption of considering classical intermediates on SACs (grey lines in the figures) could be no longer valid. The authors should check this as well. In brief, the authors should include the mentioned data to support their assumption, i.e. that the fully unconventional mechanism is the likely one in all or most of the cases.

1.3 The authors constructed scaling relationships based on a single matrix, i.e. the metal atoms anchored on the same support. It is known that the support play an important role, even as important as that of the metal [10.1002/cctc.202200611]. Given the broad impact of the journal, one could expect some universality in the provided relationships. In this respect, I encourage the authors to include results from a different support, e.g. defective graphene in place of N-dopants.

I would like to point out, that the inclusion of the additional results could boost the impact of

the authors' findings, providing a generality to the already interesting message. At the current stage, the message is nice, but it could be restricted to a specific assumed mechanism and supporting matrix.

2) The overpotential calculation does not account for the barrier for O₂ desorption, I guess because it is not an electrochemical process. However, if O₂^{*} is largely stable, the barrier for its desorption could be higher than the η . Did the authors consider this? Figure S5 shows many cases where O₂^{*} is strongly stable.

3) Figure 6 and the related discussion look obscure to me. They state that: "Fig. 6a shows that the screened highly active DACs prefer second OH adsorption (except for CoCu) over competitive OH deprotonation, suggesting that these DACs are selective for OCM rather than AEM"

If I read correctly the figure in several cases, about half, O^{*} formation is more favorable than OH^{*}OH^{*} one. This point is related to point 1.2.

Also, they suggest some correlation between M1-M2 distance and the overpotential, figure 6b, but if I read correctly the figure, the data are broadly scattered.

4) The section devoted to Bader charge analysis, charge density plots etc should be moved to the SI. They make more diluted the main findings.

5) Quality of figure 3 must be improved.

6) Some info on the peroxo or superoxo nature of the O₂^{*} complex should be provided. A possible descriptor is the O-O distance [[10.1021/acscatal.2c03020].

Reviewer #2 (Remarks to the Author):

In this manuscript, a large-scale DFT was employed to explore the feasibility of *O-*O coupling mechanism, which can circumvent the scaling relationship on improving the OER performance of DACs. In addition, the chemical and structural origin in favor of *O-*O coupling mechanism thus leading to enhanced OER activities have been revealed. However, the analyses of Gibbs free energy and overpotential are probably wrong, the conclusions lack innovative. The reviewer suggests that if the author thoroughly revises this manuscript, it may be reconsidered for publication in this journal.

(1) Equations (25) and (26) are probably wrong, and it is not appropriate to include ΔG_4 in the equation ($\eta = \max [\Delta G_1, \Delta G_2, \Delta G_3, \Delta G_4]/e - 1.23 \text{ V}$) for overpotential. In particular, the reaction-limiting step of many DACs selected by the initial analysis is *OH-*O→O^{*}-O^{*}, and it is wrong to consider $\Delta G_4/e - 1.23 \text{ V}$ as the overpotential. The author needs to do an in-depth study of the references [Nat. Commun. 11, 4066 (2020). Nat. Energy 4, 329-338 (2019). Nat. Commun. 12, 5589 (2021).].

(2) Unlike the referenced of DACs [J. Mater. Chem. A 10, 8309-8323 (2022).], in this work, the first *OH is adsorbed in the middle of the dual-atom, then the adsorption behavior of the second *OH has a stereo-hindrance effect. Therefore, the different adsorption behavior of the second *OH should be discussed, e.g. on the other side of the catalyst, on the adjacent C atom, etc.

(3) The reason for proposing LOM or even OCM is the fact that *O-*O structures are generally more stable than OO^{*} structures. Firstly, many of the selected DACs do not have this property, and secondly, the energy barrier of the step *O-*O→O₂ is generally higher, therefore, the author needs to calculate the transition state, or even the kinetic pathways of the OCM.

(4) The mechanism, OCM, that the author summarized, applies to both acidic and alkaline electrolytes. However, like the AEM, the potential-determining step is generally different for

acidic and alkaline electrolytes, and the author needs to add discussions based on the calculation results.

(5) It is necessary to carefully verify the rightness of the data for the OCM of NiPd@NC, because, the data do not theoretically match the given structures and mechanism. If the data are verified to be right, the author needs to analyze the electronic properties such as Bader charge for each step of the OCM of NiPd@NC.

Responses for the Nature Communications

Manuscript ID: NCOMMS-22-48108

Title: Synergy of Dual-atom Catalysts Deviated from the Scaling Relationship for the Oxygen Evolution Reaction

At the outset, we would like to thank the Editor for providing us with the opportunity to revise the manuscript. We are also very grateful to the reviewers for their time as well as valuable comments and suggestions. Provided below is our detailed point-to-point response to each question. The explanations to the comments from reviewers are shown in blue color. The modifications provided in the revised manuscript and Supplementary Information are highlighted with yellow background.

Responses to the comments of reviewers:

Reviewer #1 (Remarks to the Author):

In this work the authors presented a computational study on the catalytic activity of dual atom catalysts for the electro-chemical oxygen evolution reaction, finding some recipes for the catalyst optimization, that could be of help for both the theoretical and experimental design of novel materials.

I must say that current level of investigation, and generality of the message is not sufficient for Nature Communications. If the author could provide further and broader evidence of the following major points, I will be happy to recommend publication of the work.

Response:

We really appreciate the reviewer's efforts on reviewing our manuscript and providing us with constructive comments and suggestions to further improve the quality of our paper. All the suggestions and comments have been carefully addressed and modifications have been made accordingly.

Comment 1: The take-home message is unclear, and this strongly harm the potential impact of the work. More specifically, it is unclear whether the main message is one or a combination between the

following:

1.1 OER does not follow the typical mechanism.

1.2 The authors are providing a recipe for the catalytic optimization, with a volcano top different from that of conventional catalysts.

1.3 New scaling relationships are provided for OER on DACs.

Each of the mentioned points should be better corroborated by additional calculations or acknowledgement of the existing literature.

Response:

Many thanks for the reviewer's comment. The take-home message we are aiming to present in this manuscript is the potential and feasibility of *O-*O coupling mechanism (OCM) in improving the OER activity on the dual-metal sites. In addition to other paths reported in literatures, we proposed that OCM should also be considered as a possible mechanism, especially when dealing with DACs systems (point 1.1). The more favorable configuration of two *OH adsorption on the M₂ dimer can tactfully circumvent the scaling relationship between *OOH and *OH, rendering η closer to the ideal 0 V with significantly enhanced OER activities (point 1.3). More importantly, based on the large-scale screening, we further proposed the chemical and structural conditions for the feasible occurrence of OCM, thus providing a recipe for the catalytic optimization (point 1.2).

Therefore, these three aspects are in progressive relation, which construct a new framework on improving the OER activities of DACs systems via unconventional OCM pathway. To strengthen the main purpose of our manuscript, the last Discussion part has been modified in the revised manuscript.

Revisions:

Page 19, Line 20:

“This work provides a comprehensive understanding of the OCM mechanism, rendering a new recipe for the catalytic optimization of OER, with a volcano top different from that of conventional AEM. It is important that OCM should be considered as a prominent OER pathway catalyzed by DACs in the future study. These theoretical insights would be also beneficial in designing high-performance catalysts for other electrochemical reactions.”

Comment 1.1: The message is not totally new, since it has been recently demonstrated that OER could follow the alternative path proposed by the authors although with just a single metal center (single-atom catalysts) [10.1016/j.jcat.2022.12.014; 10.1021/acscatal.2c03020; 10.1021/acscatal.0c00815]. The fact that SACs can form molecular complexes, as coordination chemistry compounds is a very important hallmark of this class of materials, and it is actually more general than OER [10.1002/smt.201800368; 10.1021/jacs.1c10470]. The mentioned literature should be acknowledged.

Response:

We thank the reviewer for letting us know these excellent works, wherein new mechanisms were reported as the complement of traditional ones. Similar with these literatures, also inspired by the lattice oxygen mechanism (LOM) that involved two active sites on metal oxide, we want to know if the *O-*O coupling mechanism (OCM) could be operative on two metal centers of DACs. Although the adsorption of two *OH has already been reported on SACs, the importance and generality of their formation on DACs along with their contribution to OER activities have not systematically investigated crossing the entire periodic table so far. Furthermore, no such large-scale DFT screening to explore the feasibility of OCM with the construction of an activity map on DACs has been reported.

The literatures that reviewer mentioned help us deepen the mechanistic understanding of OER on SACs, and we further extended this concept to DACs. All these literatures have been acknowledged in the revised manuscript.

Revisions:

Page 4, Line 11:

“Recent studies have proposed the possible existence of non-conventional intermediates on SACs, for instance, the formation of dihydride in HER,⁵³ two *OH species in ORR and OER,^{54,55} as well as superoxo and peroxo complexes in OER,⁵⁶ all of which could alter the traditional reaction pathways leading to significantly different reactivities.”

References, Page 27:

53. Di Liberto, G., Cipriano, L. A., Pacchioni, G., Role of dihydride and dihydrogen complexes in hydrogen evolution reaction on single-atom catalysts. *J. Am. Chem. Soc.* **143**, 20431-20441 (2021).

54. Zhong, L., Li, S., Unconventional oxygen reduction reaction mechanism and scaling relation on single-atom catalysts. *ACS Catal.* **10**, 4313-4318 (2020).
55. Barlocco, I., Cipriano, L. A., Di Liberto, G., Pacchioni, G., Does the oxygen evolution reaction follow the classical OH*, O*, OOH* path on single atom catalysts? *J. Catal.* **417**, 351-359 (2023).
56. Cipriano, L. A., Di Liberto, G., Pacchioni, G., Superoxo and peroxy complexes on single-atom catalysts: Impact on the oxygen evolution reaction. *ACS Catal.* **12**, 11682-11691 (2022).

Comment 1.2: The authors should better corroborate the findings. More specifically, I am missing if the authors checked if the unconventional intermediates are more stable than the classical ones. This is an essential point, given that it is likely than the reaction will follow a reaction channel based on the most stable intermediates. Also, the picture can be more complex, with the crossing between conventional and unconventional intermediates [10.1016/j.jcat.2022.12.014]. This could affect the scaling relationships plots, since one should consider only the most stable intermediate for each electron transfer. This also means that the assumption of considering classical intermediates on SACs (grey lines in the figures) could be no longer valid. The authors should check this as well. In brief, the authors should include the mentioned data to support their assumption, i.e. that the fully unconventional mechanism is the likely one in all or most of the cases.

Response:

Thank you for these insightful comments and suggestions. We agree with the reviewer that the reaction will follow a channel based on the most stable intermediates. Indeed, we noticed the critical role of the most stable intermediates performed in the reaction pathway. Comparing $\Delta G_{*OH-*OH} - \Delta G_{*OH}$ with $\Delta G_{*O} - \Delta G_{*OH}$ can determine the more readily generated intermediates. As shown in Figure R1a and Tables R1-R5, it can be found that about half of the DACs exhibited $*OH-*OH$ selectivity, including 6 heteronuclear and 4 homonuclear catalysts screened. In addition, further exploration of catalysts with $*OH-*OH$ selectivity showed that $*OH-*OH$ intermediates were more readily generated as $*OH-*O$ rather than $*OOH$ (Figure R1b and Tables R6-R10). This is because the $*OH-*O$ configuration is energetically more stable than $*OOH$ on the metal dimer structure, which is distinct from SACs (Figure R2). Since 10 of the screened 11 DACs (except CoCu) possessed $*OH-*OH$ and $*OH-*O$ selectivity, crossing of intermediates between AEM and OCM can be excluded to affect the reactivity (Figure R3).

Regarding the reviewer's concern, we conducted additional calculations. For the DACs with $*O$ selectivity (Figure R1a, the pink area), the intricate case of crossing intermediates between OCM and AEM induces the reaction to follow a hybrid pathway ($*OH \rightarrow *O \rightarrow *OH-*O \rightarrow *O_2 \rightarrow O_2(g)$). As an example, for DACs with $*O$ selectivity shown in Figure R4 (selected as $\Delta G(*OH-*O) > 0$, yellow highlighted in Tables R1-R4), the potential determining step (PDS) of all DACs except CoAu is $*OH-*O \rightarrow *O_2$, and hybrid mechanism for crossing intermediates are only effective for $\Delta G(*OH \rightarrow *O)$ and $\Delta G(*O \rightarrow *OH-*O)$, therefore with limited effect on the catalytic activity of DACs with $*O$ selectivity.

Remarkably, the PDS of CoAu didn't occur at $*\text{OH}-*\text{O} \rightarrow *\text{O}_2$, which enabled the reaction to bypass the PDS of $*\text{OH} \rightarrow *\text{OH}-*\text{OH}$ ($\Delta G=1.88\text{eV}$), reducing the PDS to 1.25eV (Figure R4c). This indicates that the increase in OER reactivity is feasible for CoAu via the crossing between OCM and AEM intermediates. Nevertheless, as the PDS for the majority of DACs with $*\text{O}$ selectivity corresponds to $*\text{OH}-*\text{O} \rightarrow *\text{O}_2$, and the hybrid pathway only contributes to $\Delta G(*\text{OH} \rightarrow *\text{O})$ and $\Delta G(*\text{O} \rightarrow *\text{OH}-*\text{O})$, there is limited impact on the critical scaling relationship between $\Delta G_{*\text{OH}}$ and $\Delta G_{*\text{OH}-*\text{O}}$.

As for the grey line in Figure 4b in the manuscript, it is worth mentioning that other paths are also possible for OER on SACs. These imply the formation of other intermediates than those classical ones, like for instance in the ref [10.1016/j.jcat.2022.12.014]. However, for the purpose of this work, that is elaborating the essential role of the unconventional OCM path on DACs mediated OER, considering the most common traditional path on SACs is sufficient. For more convincing, we further added the data that shows (Figure R5), not only SACs, but also DACs and metal oxides can hardly break the 0.37V bottleneck, as long as the conventional pathway operated by the grey line.

In summary, by comparing the OCM and AEM intermediates to determine the selectivity of the reactions, it was found that 10 of the 11 high-performance DACs screened completely proceeded with the OCM along the OER process. A minimum overpotential of 0.02 V for CoAu can be achieved due to the crossing of the AEM and OCM reaction intermediates along the hybrid mechanism. The remaining DACs along the hybrid mechanism all showed limited effects on reactivity and all reaction paths (OCM, AEM and hybrid mechanism) always follow the most stable channel. Furthermore, given that the mixing mechanism only operates for $\Delta G(*\text{OH} \rightarrow *\text{O})$ and $\Delta G(*\text{O} \rightarrow *\text{OH}-*\text{O})$, the scaling relationship between the $\Delta G_{*\text{OH}}$ and $\Delta G_{*\text{OH}-*\text{O}}$ is considered to remain valid, and the grey line attributed to conventional AEM plays the crucial role in understanding the origin of the breakthrough of the 0.37 V bottleneck via OCM.

Figure R1. (a) Gibbs free energy change (ΔG) of $*OH$ toward $*OH-*OH$ versus $*OH \rightarrow *O + H^+ + e^-$. (b) Gibbs free energy change (ΔG) of $*OH-*OH \rightarrow *OH-*O + H^+ + e^-$ versus $*OH-*OH \rightarrow *OOH + H^+ + e^-$.

Figure R2. Relationship between ΔG_{*OH} and ΔG_{*OHO} for $M@NC$.

Figure R3. Gibbs free energy diagrams of AEM and OCM for comparison as well as the crossing of potential reaction paths.

Figure R4. Free energy diagram for partially *O selective M'M@NC.

Figure R5. Relationship between ΔG_{*OH} and ΔG_{*OOH} for M@NC, M'M@NC and rutile MO_2 (M = Ti, V, Ru, Rh, Ir and Pt).

Table R1. Gibbs free energy change (ΔG) for *OH toward *OH-*OH versus *OH \rightarrow *O + H⁺ + e⁻ of FeM.

FeM	$\Delta G(*OH \rightarrow *OHOH)$	$\Delta G(*OH \rightarrow *O)$
FeTi	-0.64	-0.24
FeV	-0.57	-0.14
FeCr	-0.07	-0.18
FeMn	0.19	0.07
FeFe	0.18	0.00
FeCo	0.86	0.75
FeNi	0.77	0.40
FeCu	0.60	0.83
FeZr	-0.68	0.17
FeNb	-1.05	-0.11
FeMo	-1.06	-1.18
FeRu	0.44	0.05
FeRh	0.61	0.19
FePd	0.91	0.72
FeAg	0.81	0.91
FeHf	-0.83	0.12
FeTa	-1.23	-0.18
FeW	-1.47	-0.77
FeRe	0.39	-0.72
FeOs	0.39	-0.09
FeIr	0.87	0.26
FePt	1.14	0.81
FeAu	1.13	0.77

Table R2. Gibbs free energy change (ΔG) for *OH toward *OH-*OH versus *OH \rightarrow *O + H⁺ + e⁻ of CoM.

CoM	$\Delta G(*OH \rightarrow *OHOH)$	$\Delta G(*OH \rightarrow *O)$
CoTi	-0.38	-0.12
CoV	-0.18	-0.22
CoCr	0.23	0.10
CoMn	0.19	0.03
CoCo	1.05	0.40
CoNi	0.88	0.94
CoCu	1.41	0.81
CoZr	-0.47	0.43
CoNb	-0.22	0.55
CoMo	-0.02	-0.46
CoRu	0.27	0.10
CoRh	1.12	0.46
CoPd	1.16	1.36
CoAg	0.72	0.85
CoHf	-0.65	0.37
CoTa	-0.32	0.42
CoW	0.07	-0.51
CoRe	0.01	-0.47
CoOs	0.22	-0.09
CoIr	1.11	0.35
CoPt	1.30	0.77
CoAu	1.88	1.00

Table R3. Gibbs free energy change (ΔG) for *OH toward *OH-*OH versus *OH \rightarrow *O + H⁺ + e⁻ of NiM.

NiM	$\Delta G(*OH \rightarrow *OHOH)$	$\Delta G(*OH \rightarrow *O)$
NiTi	0.03	0.37
NiV	0.17	-0.29
NiCr	0.23	0.19
NiMn	0.48	0.82
NiNi	1.28	0.49
NiCu	0.92	0.98
NiZr	-0.54	0.19
NiNb	-0.74	0.41

NiMo	-0.04	-0.66
NiRu	0.29	0.42
NiRh	1.04	0.67
NiPd	0.88	1.27
NiAg	0.80	1.30
NiHf	-0.59	0.24
NiTa	-0.92	0.42
NiW	-0.27	-0.70
NiRe	-0.19	-0.73
NiOs	-0.06	0.32
NiIr	0.85	0.57
NiPt	1.11	1.29
NiAu	1.00	1.42

Table R4. Gibbs free energy change (ΔG) for *OH toward *OH-*OH versus *OH \rightarrow *O + H⁺ + e⁻ of CuM.

CuM	$\Delta G(*OH \rightarrow *OHOH)$	$\Delta G(*OH \rightarrow *O)$
CuTi	-0.93	-0.53
CuV	0.05	-0.46
CuCr	0.33	0.21
CuMn	0.38	0.72
CuCu	1.48	2.00
CuZr	-0.53	0.33
CuNb	-0.63	-0.42
CuMo	-0.63	-0.04
CuRu	0.25	0.33
CuRh	1.05	1.02
CuPd	1.10	1.16
CuAg	1.28	1.65
CuHf	-0.72	0.33
CuTa	-0.92	-0.44
CuW	-0.88	-0.93
CuRe	-0.49	-0.87
CuOs	-0.02	0.03
CuIr	0.73	0.94
CuPt	1.21	1.67
CuAu	1.53	1.44

Table R5. Gibbs free energy change (ΔG) for *OH toward *OH-*OH versus *OH \rightarrow *O + H⁺ + e⁻ of M₂.

M ₂	$\Delta G(*OH \rightarrow *OHOH)$	$\Delta G(*OH \rightarrow *O)$
Ti₂	-0.80	-0.56
V₂	-0.33	-0.26
Cr₂	-1.30	-0.91
Mn ₂	-0.21	-0.30
Zr₂	-1.08	-0.23
Nb₂	-0.82	-0.16
Mo ₂	-0.85	-0.86
Ru ₂	0.31	0.16
Rh ₂	1.13	0.36
Pd₂	0.75	0.75
Ag₂	1.66	1.84
Hf₂	-1.41	-0.38
Ta₂	-1.36	-0.39
W₂	-1.32	-1.09
Re₂	-0.34	-0.23
Os ₂	0.28	-0.05
Ir ₂	1.23	0.04
Pt₂	0.68	0.73
Au₂	0.58	0.97

Table R6. Gibbs free energy change (ΔG) for $*OH-*OH \rightarrow *OH-*O + H^+ + e^-$ versus $*OH-*OH \rightarrow *OOH + H^+ + e^-$ of FeM.

FeM	ΔG	ΔG
	(*OHOH \rightarrow *OHO)	(*OHOH \rightarrow *OOH)
FeTi	-0.37	3.75
FeV	0.14	/
FeCu	0.58	2.53
FeZr	-0.44	4.24
FeNb	-0.22	/
FeAg	0.62	2.57
FeHf	-0.37	4.36
FeTa	-0.38	5.03
FeW	-0.18	5.39

Table R7. Gibbs free energy change (ΔG) for $*OH-*OH \rightarrow *OH-*O + H^+ + e^-$ versus $*OH-*OH \rightarrow *OOH + H^+ + e^-$ of CoM.

CoM	ΔG	ΔG
	(*OHOH \rightarrow *OHO)	(*OHOH \rightarrow *OOH)
CoTi	-0.44	/
CoNi	0.54	2.58
CoZr	-0.38	3.96
CoNb	-0.34	3.28
CoPd	0.96	1.95
CoAg	1.34	3.07
CoHf	-0.25	4.21
CoTa	-0.33	/

Table R8. Gibbs free energy change (ΔG) for $*OH-*OH \rightarrow *OH-*O + H^+ + e^-$ versus $*OH-*OH \rightarrow *OOH + H^+ + e^-$ of NiM.

NiM	ΔG	ΔG
	(*OHOH \rightarrow *OHO)	(*OHOH \rightarrow *OOH)
NiTi	-0.24	3.16
NiMn	0.31	/
NiCu	1.48	2.20
NiZr	0.07	3.87

NiNb	0.14	/
NiRu	0.79	/
NiPd	1.10	2.24
NiAg	1.81	2.62
NiHf	-0.01	3.74
NiTa	0.24	/
NiOs	0.73	/
NiPt	0.94	2.14
NiAu	1.74	1.91

Table R9. Gibbs free energy change (ΔG) for $*OH-*OH \rightarrow *OH-*O + H^+ + e^-$ versus $*OH-*OH \rightarrow *OOH + H^+ + e^-$ of CuM.

CuM	ΔG	ΔG
	($*OHOH \rightarrow *OHO$)	($*OHOH \rightarrow *OOH$)
CuTi	-0.07	1.33
CuMn	0.63	2.89
CuCu	1.34	1.71
CuZr	0.35	3.95
CuNb	-0.24	/
CuMo	0.10	/
CuRu	0.94	2.50
CuPd	1.25	2.00
CuAg	1.64	2.10
CuHf	0.49	4.73
CuTa	-0.27	/
CuOs	0.73	/
CuIr	1.14	2.79
CuPt	1.04	1.90

Table R10. Gibbs free energy change (ΔG) for $*OH-*OH \rightarrow *OH-*O + H^+ + e^-$ versus $*OH-*OH \rightarrow *OOH + H^+ + e^-$ of M_2 .

M_2	ΔG	ΔG
	($*OHOH \rightarrow *OHO$)	($*OHOH \rightarrow *OOH$)
Ti ₂	-0.77	4.57
V ₂	-0.61	/
Cr ₂	0.14	/

Zr ₂	-1.61	4.45
Nb ₂	-0.72	4.22
Pd ₂	1.44	2.32
Ag ₂	1.70	1.78
Hf ₂	-1.80	4.94
Ta ₂	-0.79	/
W ₂	-0.74	/
Re ₂	-0.58	/
Pt ₂	1.53	2.34
Au ₂	1.29	2.50

Revisions:

Page 13, Line 26:

“Meanwhile, the dual metal active site followed the scaling relationship significantly deviated from that of the extremely challenging AEM followed in Fig. 4b, without the constant difference of 3.2 ± 0.2 eV, which implicated that the minimum theoretical overpotential of 0.37 V was remarkably lowered by the dual metal active site, especially NiPd (0.06 V), CuPd (0.07 V) and CuPt (0.08 V).”

Page 16, Line 9:

“Here, critical questions are raised: is OCM suitable for all dual metal active sites? What is the condition that triggers the occurrence of OCM? Whether the crossings of two different mechanisms happen? To answer these questions, 105 DACs, including homonuclear composition, were explored in detail. The competition between second *OH adsorption and *OH deprotonation distinguished the two mechanisms, which was first assessed by comparing Gibbs free energy change. Fig. 6a and Supplementary Tables 70–74 show that 6 heteronuclear and 4 homonuclear highly active DACs screened prefer second *OH adsorption (except for CoCu) over competitive *OH deprotonation, suggesting that these DACs are selective for *OH-*OH rather than *O. Subsequently, based on a recent study,⁵⁵ DACs with *OH-*OH selectivity were further analyzed to evaluate the possibility of intermediate crossover (*OH-*OH→*OH-*O or *OH-*OH→*OOH). Fig. 6b and Supplementary Tables 75–79 show that all DACs with *OH-*OH selectivity proceed along the OCM without *OOH generation. This is because the *OH-*O conformation is energetically more stable than *OOH on the metal dimer structure, which explains

the scaling relationship of OCM deviating from AEM in Fig. 4b. Notably, the absence of synergistic effects in the active center of SACs caused a different scaling relationship between *OH and *OH-*O along the OCM, without overlapping with that of DACs (Supplementary Fig. 15), which resulted in a complicated OER reaction network of the SACs.”

Page 16, Line 25:

“For the DACs with *O selectivity in Fig. 6a, the intricate case of crossing intermediates between OCM and AEM induces the reaction to follow a hybrid pathway (*OH→*O→*OH-*O→*O₂→O₂(g)). For the examples of DACs with *O selectivity shown in Supplementary Fig. 16 (selected as $\Delta G(*OH-*O) > 0$, Supplementary Tables 70–73), the PDS of all DACs except CoAu is *OH-*O→*O₂, and hybrid mechanism for crossing intermediates are only effective for $\Delta G(*OH→*O)$ and $\Delta G(*O→*OH-*O)$, therefore with limited effect on the catalytic activity of DACs with *O selectivity. Remarkably, the PDS of CoAu didn’t occur at *OH-*O → *O₂, which enabled the reaction to bypass the PDS of *OH → *OH-*OH ($\Delta G = 1.88\text{eV}$), reducing the PDS to 1.25eV (Supplementary Fig. 16c). This indicates that the increase in OER reactivity is feasible for CoAu via the crossing between OCM and AEM intermediates.”

Page 17, Line 9:

Figure R1 has been included as Fig.6 in the revised manuscript.

Supplementary Figure and Table:

Figure R2 has been included in Supplementary Fig. 15.

Figure R3 has been included in Supplementary Fig 4.

Figure R4 has been included in Supplementary Fig. 16.

Figure R5 has been included in Supplementary Fig. 11.

Tables R1-R10 have been included in Supplementary Tables 70-79.

Comment 1.3: The authors constructed scaling relationships based on a single matrix, i.e. the metal atoms anchored on the same support. It is known that the support plays an important role, even as important as that of the metal [10.1002/cctc.202200611]. Given the broad impact of the journal, one

could expect some universality in the provided relationships. In this respect, I encourage the authors to include results from a different support, e.g. defective graphene in place of N-dopants.

I would like to point out, that the inclusion of the additional results could boost the impact of the authors' findings, providing a generality to the already interesting message. At the current stage, the message is nice, but it could be restricted to a specific assumed mechanism and supporting matrix.

Response:

We thank the reviewer for the constructive and inspiring comments. We fully agree that support also plays a key role in electrochemical reactions, as indicated in the literature [10.1002/cctc.202200611]. Accordingly, the calculation of defective graphene as a substrate to support metal diatoms was supplemented. The prototype was constructed (Figure R6) based on the experimental synthesis (*Nano Lett.* **2014**, 14, 3766-3772) and the corresponding computational study (*ACS Catal.* **2015**, 5, 6658-6664).

The activity trends of 86 M'M@Gr along the OCM for OER are shown as heat map in Figure R7, with all oxygenated intermediates as the most stable conformation (Tables R11-R14). Analogous to Fig.2b in the manuscript, the heat map also demonstrates clearly from left to right that the blue area representing the high overpotential gradually decreases, indicating that the early-transition metal doping is unfavorable for OER performance in either N-doped or defective graphene. In contrast to Fig. 2b, the quantities of DACs with high activity on defective graphene support are much less than that on N-doped graphene support, following the order $\text{FeFe}(0.38\text{V}) = \text{CuPd}(0.38\text{V}) < \text{CoAg}(0.39\text{V}) < \text{NiCu}(0.40\text{V}) < \text{CoPd}(0.42\text{V}) < \text{CoCu}(0.43\text{V}) = \text{CuCu}(0.43\text{V}) < \text{NiAu}(0.50\text{V})$. Based on the analysis of the free energy diagrams (Figure R8), the above 8 M'M@Gr exhibited strong adsorption of oxygenated intermediates, with consequently high overpotential and difficulty in O₂ desorption, thus unfavorable for the sustainability of reaction.

Notably, although the OER activity by OCM path on M'M@Gr was inferior, the critical intermediates follow the same scaling relationship in Fig. 4b, albeit with a different support (Figure R9), indicating a general rule of OCM on DACs.

Figure R6. Optimized structure of M'M@Gr monolayer in a $p(6 \times 6)$ supercell.

Figure R7. The computed overpotential (η) of the FeM, CoM, NiM, and CuM dimers on M'M@Gr.

Figure R8. Gibbs free energy diagram along the OCM of M'M@Gr.

Figure R9. Relationship between ΔG^*_{OH} and ΔG^*_{OH-*O} for M'M@Gr.

Table R11. Adsorption free energy of FeM@Gr for *OH and the calculated relative energy of the optimized adsorption configuration for *OH-*OH. Bold fonts represent the most stable configuration.

FeM						
FeTi	-1.92	/	/	/	0.00	/
FeV	-2.03	/	-1.43	/	0.00	0.63
FeCr	-2.00	/	-1.40	0.00	0.09	/
FeMn	-1.95	/	/	0.00	0.20	/
Fe ₂	-1.55	-1.12	/	0.00	/	/
FeCo	-1.22	-1.16	/	0.00	/	/
FeNi	-1.06	-1.11	/	0.00	0.20	/
FeCu	-0.85	/	-0.74	0.07	/	0.00
FeZr	-1.89	/	/	0.90	0.00	
FeNb	-1.87	/	-1.37	0.75	0.00	
FeMo	-1.59	/	-1.37		0.00	0.95
FeRu	-1.32	-1.08	-0.95	0.00	/	0.52
FeRh	-1.12	-1.20	-0.48	0.00	/	0.41
FePd	-1.03	-1.15	/	0.00	/	0.29
FeAg	-2.09	-2.17	/	0.00	/	0.14
FeHf	-1.99	/	-1.52	/	0.00	1.15
FeTa	-2.04	/	-1.72	/	0.00	/
FeW	-1.80	/	-1.88	0.27	0.00	/
FeRe	-1.48	/	-1.77	0.18	0.00	/
FeOs	-1.49	-1.07	-1.55	0.02	0.00	/
FeIr	-1.21	-1.15	-0.96	0.00	0.23	/
FePt	-1.05	-1.12	-0.43	0.00	/	/
FeAu	-0.87	-0.81	0.01	/	0.29	0.00

Table R12. Adsorption free energy of CoM@Gr for *OH and the calculated relative energy of the optimized adsorption configuration for *OH-*OH. Bold fonts represent the most stable configuration.

CoM						
CoTi	-1.81	/	-1.26	/	0.00	/
CoV	-1.71	/	-1.38	/	0.00	0.86
CoCr	-1.48	/	-1.31	0.15	0.00	/
CoMn	-1.56	/	-1.33	0.00	0.07	/
Co ₂	-1.20	/	/	0.00	/	/
CoNi	-0.94	-0.64	/	/	0.33	0.00
CoCu	-0.46	-0.46	/	/	0.35	0.00
CoZr	-1.76	/	-1.16	0.87	0.00	/
CoNb	-1.56	/	-1.38	0.98	0.00	/
CoMo	-1.22	/	-1.28	/	0.00	/
CoRu	-1.03	/	-1.00	0.00	/	/
CoRh	-1.00	/	-0.56	0.00	/	/
CoPd	-0.84	/	/	0.11	/	0.00
CoAg	-0.61	-0.54	/	/	0.24	0.00
CoHf	-1.93	/	-1.47	1.41	0.00	/
CoTa	-1.76	/	-1.76	/	0.00	/
CoW	-1.46	/	-1.80	/	0.00	/
CoRe	-1.16	/	-1.62	/	0.00	/
CoOs	-1.14	/	-1.55	0.00	0.01	/
CoIr	-1.11	-0.84	-1.05	0.00	/	/
CoPt	-0.94	-0.67	/	/	0.00	/
CoAu	-0.56	-0.39	/	/	0.05	0.00

Table R13. Adsorption free energy of NiM@Gr for *OH and the calculated relative energy of the optimized adsorption configuration for *OH-*OH. Bold fonts represent the most stable configuration.

NiM						NiTi	-1.54	/	-1.17	/	0.00	/
NiV	-1.24	/	-1.27	/	0.00	/
NiCr	-1.08	/	-1.27	0.18	0.00	/
NiMn	-0.92	/	-1.31	0.06	0.00	/
Ni ₂	-0.63	/	-0.03	/	0.00	/
NiCu	-0.50	/	/	/	/	0.00
NiZr	-1.54	/	-1.08	0.93	0.00	/
NiNb	-1.14	/	-1.29	/	0.00	/
NiMo	-0.57	/	-1.08	/	0.00	/
NiRu	-0.80	/	/	0.00	/	/
NiRh	-0.71	/	-0.35	0.00	/	0.46
NiPd	-0.55	/	0.06	/	0.00	0.25
NiAg	-0.42	/	/	/	0.00	0.95
NiHf	-1.68	/	-1.44	/	0.00	/
NiTa	-1.34	/	-1.69	/	0.00	/
NiW	/	/	-1.60	/	0.00	/
NiRe	/	/	-1.87	0.36	0.00	/
NiOs	-0.94	/	-1.50	0.03	0.00	/
NiIr	-0.87	/	-0.86	/	0.00	/
NiPt	-0.66	/	-0.31	/	0.00	0.60
NiAu	-0.61	/	/	/	0.00	0.25

Table R14. Adsorption free energy of CuM@Gr for *OH and the calculated relative energy of the optimized adsorption configuration for *OH-*OH. Bold fonts represent the most stable configuration.

CuM						CuTi	/	/	-1.56		0.00	
CuV	/	/	-1.18		0.00	
CuCr	-0.74	/	-1.41	0.28	0.00	
CuMn	-0.78	/	-1.19	0.10	0.00	
Cu2	-0.15	/	/		0.00	
CuZr	/	/	-1.21		0.00	
CuNb	/	/	-1.14		0.00	
CuMo	/	/	-1.32		0.00	
CuRu	-0.39	/	-0.76	0.00		
CuRh	-0.23	/	-0.28		0.00	
CuPd	-0.30	/	/		0.00	0.75
CuAg	-0.20	/	/		0.00	1.51
CuHf	/	/	-1.56		0.00	
CuTa	/	/	-1.57		0.00	
CuW	/	/	-1.85		0.00	
CuRe	/	/	-1.66		0.00	
CuOs	-0.49	/	-1.31	0.40	0.00	
CuIr	-0.34	/	-0.70		0.00	
CuPt	-0.50	/	-0.20		0.00	
CuAu	-0.32	/	/		0.00	0.27

Revisions:

Page 12, Line 30:

“As the support also plays a vital role in electrochemical reactions,⁸⁰ it is necessary to explore the activity of different support for the same metal dimers (FeM, CoM, NiM and CuM). Two adjacent single vacancies on graphene accommodating metal dimers (M'M@Gr) were used as the prototype^{81,82} (Supplementary Fig. 7) to further evaluate the support effect for the catalytic performance of DACs along the OCM. The activity trends of 86 M'M@Gr along the OCM for OER are shown as heat map in Supplementary Fig. 8, with all oxygenated intermediates as the most stable configuration (Supplementary Tables 57-60). In analogy to Fig. 2b, the heat map also

clearly shows a progressively improved activity from left to right, indicating that early-transition metal doping disadvantages the OER performance in either N-doped or defective graphene. In contrast to Fig. 2b, the quantities of DACs with high activity on defective graphene support are much less than that on N-doped graphene support, following the order $\text{FeFe}(0.38\text{V}) = \text{CuPd}(0.38\text{V}) < \text{CoAg}(0.39\text{V}) < \text{NiCu}(0.40\text{V}) < \text{CoPd}(0.42\text{V}) < \text{CoCu}(0.43\text{V}) = \text{CuCu}(0.43\text{V}) < \text{NiAu}(0.50\text{V})$. Based on the analysis of the free energy diagrams (Supplementary Fig. 9), the above 8 M'M@Gr exhibited strong adsorption of oxygenated intermediates, with consequently high overpotential and difficulty in O_2 desorption, thus unfavorable for the sustainability of reaction.”

Page 13, Line 30:

“Notably, despite the fact that OER activity on M'M@Gr was inferior due to the excessively strong adsorption of oxygenated intermediates, the critical intermediates follow the same scaling relationship in Fig. 4b, albeit with a different support (Supplementary Fig. 12), indicating a general rule of OCM on DACs.”

References, Page 29:

80. Di Liberto, G., Cipriano, L. A., Pacchioni, G., Single atom catalysts: What matters most, the active site or the surrounding? *ChemCatChem* **14**, 202200611 (2022).
81. Li, Y., Su, H., Chan, S. H., Sun, Q., CO_2 electroreduction performance of transition metal dimers supported on graphene: A theoretical study. *ACS Catal.* **5**, 6658-6664 (2015).
82. He, Z., *et al.*, Atomic structure and dynamics of metal dopant pairs in graphene. *Nano Lett.* **14**, 3766-3772 (2014).

Supplementary Figure and Table:

Figure R6 has been included in Supplementary Fig. 7.

Figure R7 has been included in Supplementary Fig. 8.

Figure R8 has been included in Supplementary Fig. 9.

Figure R9 has been included in Supplementary Fig. 12.

Tables R11-R14 have been included in Supplementary Tables 57-60.

Comment 2: The overpotential calculation does not account for the barrier for O₂ desorption, I guess because it is not an electrochemical process. However, if O₂* is largely stable, the barrier for its desorption could be higher than the η . Did the authors considered this? Figure S5 shows many cases where O₂* is strongly stable.

Response:

We are thankful to the reviewer for the professional comment. Given that the desorption of gaseous products during electrochemical reactions like NRR (*J. Am. Chem. Soc.* **2020**, 142, 5709–5721), CO₂RR (*J. Mater. Chem. A* **2021**, 9, 8761 – 8771) and OER (*Science*, **2016**, 352, 333-337.; *Nat. Energy*, **2019**, 4, 329-338.; *Nat. Commun.*, **2020**, 11, 4066.) does not involve electron transfer, with physical methods such as temperature, pressure and stirring can be accomplished. We agree that the high desorption barrier of O₂ is unfavorable to the reaction. With reference to our previous experimental work (*Nat. Commun.* **2021**, 12, 5589), we investigated the desorption of O₂ for FeM, CoM, NiM and CuM using the desorption free energy of NiFe-CNG as a benchmark ($\Delta G^{*O_2 \rightarrow O_2(g)} = 1.17$ eV), and found that the $\Delta G^{*O_2 \rightarrow O_2(g)}$ of all 8 highly active DACs screened was below 1.17 eV (Table R15). Figure S5 can be calculated by: $\Delta G^{*O_2} = G^{*O_2} - G_{slab} - G_{O_2(g)}$.

Table R15. Desorption free energy change ($\Delta G^{*O_2 \rightarrow O_2(g)}$) for *O₂ toward O₂ of FeM, CoM, NiM and CuM. Bolded font: With a benchmark of 1.17 eV for NiFe-CNG.

	FeM	CoM	NiM	CuM
Ti	2.07	1.74	2.54	2.70
V	2.29	1.99	3.04	2.36
Cr	2.93	2.46	1.95	2.12
Mn	2.48	2.04	1.29	1.34
Fe	2.09	1.46	1.23	1.36
Co	1.46	1.49	1.09	0.88
Ni	1.23	1.09	0.27	0.28
Cu	1.36	0.88	0.28	0.18
Zr	2.79	2.86	2.52	2.52
Nb	2.54	3.08	3.40	2.37
Mo	2.82	2.44	3.71	2.81
Ru	1.70	1.95	1.69	1.68
Rh	1.44	1.39	0.77	0.45

Pd	1.82	0.96	0.39	0.10
Ag	1.73	0.84	0.70	0.20
Hf	2.62	2.88	2.66	2.94
Ta	3.73	2.01	4.02	2.71
W	3.90	4.02	4.45	3.27
Re	2.68	2.07	2.49	3.94
Os	1.72	1.92	1.87	1.78
Ir	1.45	1.49	0.78	0.45
Pt	1.80	0.88	0.26	0.15
Au	1.46	0.81	0.17	-0.11

Revisions:

Page 12, Line 11:

“It is worth noting that $*O_2 \rightarrow O_2(g)$, as a non-electrochemical process,⁷⁹ is not appropriate to be included in the elementary steps, which overestimates the overpotential and misjudges the PDS.”

Page 12, Line 15:

“Supplementary Fig. 2 illustrates the specific adsorption configurations which are indicated in Fig. 3 as the peroxo- or superoxide nature of $*O_2$. On the other hand, the desorption of $*O_2$ was also evaluated for 86 DACs with a benchmark value of an experimentally reported DACs that follows OCM ($\Delta G_{*O_2 \rightarrow O_2(g)} = 1.17\text{eV}$).⁵⁹ The excessive $\Delta G_{*O_2 \rightarrow O_2(g)}$, i.e. higher than 1.17eV, could be unfavorable for the reaction cycle, and it turns out that 8 high-performance DACs screened in the present study possess favorable desorption capacity with moderate O-affinity (Supplementary Table 56).”

Supplementary Table:

Table R15 has been included in Supplementary Table 56.

References, Page 28:

79. Zhang, B., *et al.*, Homogeneously dispersed multimetal oxygen-evolving catalysts. *Science* **352**, 333-337 (2016).

Comment 3: Figure 6 and the related discussion look obscure to me. They state that: “Fig. 6a shows that the screened highly active DACs prefer second OH adsorption (except for CoCu) over competitive OH deprotonation, suggesting that these DACs are selective for OCM rather than AEM”

If I read correctly the figure in several cases, about half, O* formation is more favorable than OH*OH* one. This point is related to point 1.2.

Also, they suggest some correlation between M1-M2 distance and the overpotential, figure 6b, but if I read correctly the figure, the data are broadly scattered.

Response:

We thank the reviewer for the comment. What we want to express is that the 7 heteronuclear and 4 homonuclear highly active DACs screened have *OH-*OH selectivity instead of *O (except for CoCu marked in a different color in Fig. 6a), and we further supplement calculations demonstrating that *OH-*OH intermediates were more readily generated as *OH-*O rather than *OOH (Figure R1b), which means OCM is preferred over AEM on these systems. With regard to the selectivity of reaction intermediate species and the possibility of crossover, please refer to the response for **Comment 1.2**.

As for the relationship between M1-M2 distance and the overpotential, we just want to emphasize that the moderate range of M1-M2 distance (2.35 ~ 2.45 Å) would benefit the occurrence of OCM. While no clear correlation between M1-M2 distance and the overpotential can be found, this is not surprising because the reactivities of OER via OCM are largely related to the electronic factors. The M1-M2 distance within this range does not necessarily means a high OER activity. To avoid confusion for the readers, we have revised the relevant discussions and moved the figure to Supplementary information.

Revisions:

Page 19, Line 5:

“We also plotted the M1-M2 distance versus η , as shown in Supplementary Fig. 22, with the appropriate range favoring the occurrence of OCM.”

Comment 4: The section devoted to Bader charge analysis, charge density plots etc should be moved to the SI. They make more diluted the main findings.

Response:

We thank the reviewer for the comments. We have moved Fig. 7 in the original manuscript to Figure S18 in the revised Supplementary information.

Revisions:

Supplementary Figure:

Fig.7 in the original manuscript has been included in Supplementary Fig.18.

Comment 5: Quality of figure 3 must be improved.

Response:

Thank you for this comment. We have modified Fig. 3 in the revised manuscript.

Comment 6: Some info on the peroxo or superoxo nature of the O₂* complex should be provided. A possible descriptor is the O-O distance [10.1021/acscatal.2c03020].

Response:

We thank the reviewer for the constructive comments. As mentioned in the literature (*ACS Catal.* **2022**, 12, 11682-11691), peroxide or superoxide nature of *O₂ complexes play an important role on SACs for OER, which could be the same on DACs. Therefore, we inspect the nature of *O₂ complexes along OER path for the highly active DACs screened, which are shown in Fig. 3 in the revised manuscript and Supplementary Fig. 12.

Furthermore, as suggested by the reviewer, we found a correlation between the charge amount of *O₂ received from the slab and the O-O bond length of *O₂ (Figure R10a), implying that the more charge *O₂ receives, the longer the bond length. Particularly, after excluding the disturbance data of DACs with $\Delta G_{*O_2 \rightarrow O_2(g)} > 1.17$ eV (see response for Comment 2), this linear correlation is more satisfactory (Figure R10b). Therefore, the O-O distance of *O₂ can be employed as a useful descriptor to describe the charge transfer. In addition, to better understand the possible relationship between the

bond length and the O₂ desorption, we also plotted the correlation with bond length and $\Delta G^{*O_2 \rightarrow O_2(g)}$. Unfortunately, no strong correlation exists between the both, which is possibly caused by the wide energy range of the $\Delta G^{*O_2 \rightarrow O_2(g)}$ (Figure R10c, d).

Figure R10. (a, b) The relationship between *O₂ bond length and the charge it obtained. (c, d) Relationship between *O₂ bond length and $\Delta G^{*O_2 \rightarrow O_2(g)}$.

Revisions:

Page 12, Line 13:

“Inspired by the literature,⁵⁶ *O₂ exhibits peroxide or superoxide characteristics in two different adsorption modes, end-on and side-on, respectively, due to the different combinations that induce variation in O-affinity. Supplementary Fig. 2 illustrates the specific adsorption configurations which are indicated in Fig. 3 as the peroxy- or superoxide nature of *O₂.”

Page 12, Line 22:

“To better understand the significance of *O_2 , a correlation between the charge amount of *O_2 received from the slab and the O-O bond length of *O_2 (Supplementary Fig. 6a) can be established, implying that the more charge *O_2 receives, the longer the bond length. Particularly, after excluding the disturbance data of DACs with $\Delta G_{^*O_2 \rightarrow O_2(g)} > 1.17$ eV, this linear correlation is more satisfactory (Supplementary Fig. 6b). Therefore, the O-O distance of *O_2 can be employed as a useful descriptor to describe the charge transfer. In addition, we also plotted the correlation with bond length and $\Delta G_{^*O_2 \rightarrow O_2(g)}$. Unfortunately, no strong correlation exists between the both, which is possibly caused by the wide energy range of the $\Delta G_{^*O_2 \rightarrow O_2(g)}$ (Supplementary Fig. 6c, d).”

Page 10, Line 10:

Fig.3 has been revised below.

Fig. 3 Gibbs free energy diagrams and optimized configurations. a-h Gibbs free energy diagrams of

NiPd@NC, CuPd@NC, CuPt@NC, CoPd@NC, NiPt@NC, NiCu@NC and the respective corresponding SACs for OER. **i** Optimized configurations of oxygenated intermediates represented as NiPd@NC.

Supplementary Figure:

Supplementary Fig.13 is revised below

Figure S13. Gibbs free energy diagrams of M₂@NC for OER.

Supplementary Figure:

Figure R10 has been included in Supplementary Fig. 6.

Reviewer #2 (Remarks to the Author):

In this manuscript, a large-scale DFT was employed to explore the feasibility of *O-*O coupling mechanism, which can circumvent the scaling relationship on improving the OER performance of DACs. In addition, the chemical and structural origin in favor of *O-*O coupling mechanism thus leading to enhanced OER activities have been revealed. However, the analyses of Gibbs free energy and overpotential are probably wrong, the conclusions lack innovative. The reviewer suggests that if the author thoroughly revises this manuscript, it may be reconsidered for publication in this journal.

Response:

Many thanks for the reviewer's comments, which provide us another chance to further improve the quality of our paper. We have carefully modified the manuscript and Supplementary Information according to your constructive comments.

Comment 1: Equations (25) and (26) are probably wrong, and it is not appropriate to include ΔG_4 in the equation ($\eta = \max [\Delta G_1, \Delta G_2, \Delta G_3, \Delta G_4]/e - 1.23 \text{ V}$) for overpotential. In particular, the reaction-limiting step of many DACs selected by the initial analysis is $*\text{OH}-*\text{O} \rightarrow \text{O}^*-\text{O}^*$, and it is wrong to consider $\Delta G_4/e - 1.23 \text{ V}$ as the overpotential. The author needs to do an in-depth study of the references [Nat. Commun. 11, 4066 (2020). Nat. Energy 4, 329-338 (2019). Nat. Commun. 12, 5589 (2021).].

Response:

We thank the reviewer for the comment and letting us know about these excellent works. First of all, in all our calculations, $*\text{OH}-*\text{O}$ deprotonation to form the adsorbed state of O_2 ($*\text{O}_2$) is employed, and $*\text{O}_2$ is equivalent to the side-on configuration ($*\text{O}-\text{O}^*$) and the end-on configuration ($*\text{OO}$). We apologise for the ambiguity and therefore we have changed the relevant expressions in the revised manuscript. Regarding Equations (26), $*\text{O}_2$ as a superoxo/peroxo complex cannot be ignored, as demonstrated in a recent literature (*ACS Catal.*, **2022**, 12, 11682-11691), also suggested by the Reviewer 1. In addition, the $*\text{O}_2$ to $\text{O}_{2(\text{g})}$ process is a non-electrochemical step (*Science*, **2016**, 352, 333-337; *Nat. Energy*, **2019**, 4, 329-338; *Nat. Commun.*, **2020**, 11, 4066), which would lead to high overpotential and unreasonable calculations if $*\text{OH}-*\text{O} \rightarrow \text{O}_{2(\text{g})}$ is set as one of the elementary steps. Therefore, the essential question is whether direct coupling to form O-O bonds occurs after $*\text{OH}-*\text{O}$ deprotonation (Equations (25)).

According to Zhang et al. (Figure R11, *Nat. Commun.*, **2020**, 11, 4066), the OER cycle on the Fe-

Ni dual-site for the (FeCoCrNi)OOH catalyst occurs following $[\text{Fe}^* + \text{Ni}^*\text{-OH}] \rightarrow [\text{Fe}^*\text{-OH} + \text{Ni}^*\text{-OH}] \rightarrow [\text{Fe}^*\text{-O} + \text{Ni}^*\text{-OH}] \rightarrow [\text{Fe}^*\text{-OO-}^*\text{Ni}]$, which means that the O-O bond is formed directly after deprotonation of $[\text{Fe}^*\text{-O} + \text{Ni}^*\text{-OH}]$. As mentioned in Supplementary Note 5, $\Delta G_4 = \Delta G_{(\text{Fe}^*\text{-OO-}^*\text{Ni})} - \Delta G_{(\text{Fe}^*\text{-O} + \text{Ni}^*\text{-OH})} - eU + \Delta G_{\text{pH}}$. In a similar study of the LOM pathway, Huang et al. reported direct O-O coupling from $[\text{OH-Co-O}]$ on $\text{Zn}_{0.2}\text{Co}_{0.8}\text{OOH}$ catalysts as well (Figure R12a, *Nat. Energy* **2019**, 4, 329-338), and the free energy calculation section of the supplementary information mentioned that $\Delta G = \Delta G_{[\text{OO+Ov}]^*} - \Delta G_{\text{OH}^*} - e^- + \Delta G_{\text{pH}}$. Furthermore, for the NiFe-CNG DACs successfully prepared by our previous work (one of authors in this work), the reaction cycle follows $[\text{Ni}^*\text{-Fe}^*] \rightarrow [* \text{OH-Ni-Fe}^*] \rightarrow [* \text{OH-Ni-Fe-}^* \text{OH}] \rightarrow [* \text{OH-Ni-Fe-}^* \text{O}] \rightarrow [* \text{O-O}^*]$ (Figure R13, *Nat. Commun.* **2021**, 12, 5589). Therefore, it is reasonable to identify the reaction limiting step for many DACs as $* \text{OH-}^* \text{O} \rightarrow ^* \text{O-O}^*$ based on the calculations and to use $\Delta G_4/e - 1.23 \text{ V}$ as the overpotential.

Additionally, we demonstrate that the two O atoms are directly coupled after $* \text{OH-}^* \text{O}$ deprotonation, rather than as two isolated $* \text{O}$ atoms. Energetically, the configuration of $* \text{O}_2$ formed by $* \text{O-}^* \text{O}$ coupling is lower and more stable than the configuration before coupling (Table R16). On the other hand, the coupling energy barrier is very low, enabling $* \text{O-}^* \text{O}$ coupling to be relatively easy (More details please see Response 1.3). In general, it is a relatively reasonable elementary step to go from $* \text{OH-}^* \text{O}$ to $* \text{O}_2$ in our work.

Figure R11. (a) Free energy diagram of OER cycling at Fe–Ni dual-site on (FeCoCrNi)OOH model.

(b) Schematic illustration of the proposed overall OER pathway for EA-FCCN catalyst. The black and blue oxygen atoms represent lattice and adsorbed oxygen, respectively. (*Nat. Commun.* **2020**, 11,

[Redacted]

Figure R12. Proposed lattice oxygen oxidation mechanism (LOM) of OER. (*Nat. Energy* **2019**, *4*, 329-338.)

Figure R13. (a, b) Free energy diagrams of NiFe-CNG models toward OER at the Ni-Fe dual site. c Schematic illustration of the proposed overall OER pathway for the NiFe-CNG catalyst. (*Nat. Commun.* **2021**, *12*, 5589.)

Table R16. Energy difference between after and before oxygen coupling, $\Delta E = E_{\text{Post-coupling}} - E_{\text{Pre-coupling}}$.

M'M@NC	$\Delta E(\text{eV})$
CoCu@NC	-0.69
CoPd@NC	-0.59
NiCu@NC	-0.42
NiPd@NC	-0.83
NiPt@NC	-0.67
CuCu@NC	-1.14

CuPd@NC	-1.55
CuPt@NC	-1.13

Revisions:

Page 17, Line 4:

“Additionally, the kinetic process of *O-*O coupling is also critical for triggering OCM, and higher kinetic barriers reduce the potential for OCM to occur. Therefore, we further calculated the possible coupling processes based on the 8 heteronuclear DACs screened (Supplementary Fig. 17 and Table 80). The results show that all 8 DACs possessed low coupling barriers and by the same magnitude as reported in the literature.⁴⁸ As a result, *O-*O coupling cannot hinder OCM from proceeding.”

Methods, Page 21:

“Where *O₂ includes the side-on configuration (*O-O*) as well as the end-on configuration (*OO), both of which employed the most stable configuration of the adsorbed O₂ in the calculations.”

Supplementary Table:

Table R16 has been included in Supplementary Table 80.

Comment 2: Unlike the referenced of DACs [J. Mater. Chem. A 10, 8309-8323 (2022).], in this work, the first *OH is adsorbed in the middle of the dual-atom, then the adsorption behavior of the second *OH has a stereo-hindrance effect. Therefore, the different adsorption behavior of the second *OH should be discussed, e.g. on the other side of the catalyst, on the adjacent C atom, etc.

Response:

Thanks for the constructive comments from the reviewer. Regarding the reviewer’s concern, we

performed additional calculations to investigate the selectivity of the different adsorption behavior of the second *OH (*J. Mater. Chem. A* 2022, **10**, 8309-8323; *J. Catal.* 2023, **417**, 351-359;). Virtually the entire DACs, the second *OH prefers to adsorb at the M-M site compared to the M-C site (Figure R14a and Tables R17-R20), which circumvents the effect of the adjacent C site for *OH adsorption. Likewise, we also investigated the possibility of the second *OH adsorption on the other side of the DACs, which can be achieved for nearly half of the 86 DACs (Figure R14b and Table R17). However, although the above calculations indicate that the possibility of *OH adsorption on the reverse side, we used a monolayer model for DACs in this work, whereas experimentally synthesized graphene-based catalysts usually consist of multilayers (*J. Mater. Chem. A*, **2020**, 8, 15809–15815), and not only intercalation effects need to be considered, such structures also prevent O-O coupling (*J. Catal.* **2023**, 417, 351–359). On the other hand, different combinations of metal dimer have various O affinities without assurance that the *OH on the other side will not undergo further evolution (e.g., *O and *OOH), which complicates the picture and is not considered in this work.

Figure R14. (a) Selectivity of the second *OH adsorption at the M-M site versus the M-C site. (b) Selectivity of the second *OH adsorption at the M-M site versus the M-M(anti) path.

Table R17. $\Delta G_{*OH \rightarrow *OH-*OH}$ of the second *OH adsorbed at different sites (M-M site, M-C site and M-M(anti) site) on the FeM@NC DACs.

FeM	$\Delta G_{*OH-*OH} - \Delta G_{*OH}$		
	M-M site	M-C site	M-M(anti) site
FeTi	-0.64	1.07	-0.61

FeV	-0.57	1.33	-0.52
FeCr	-0.07	1.64	-0.33
FeMn	0.19	1.25	-0.02
FeFe	0.18	1.53	0.29
FeCo	0.86	2.09	1.41
FeNi	0.77	1.43	0.34
FeCu	0.60	1.55	0.22
FeZr	-0.68	1.60	0.07
FeNb	-1.05	0.71	-1.04
FeMo	-1.06	0.56	-0.93
FeRu	0.44	1.27	0.17
FeRh	0.61	1.43	0.82
FePd	0.91	1.57	1.10
FeAg	0.81	1.92	1.03
FeHf	-0.83	1.25	-0.62
FeTa	-1.23	0.54	-1.11
FeW	-1.47	0.43	-1.41
FeRe	0.39	1.32	-0.19
FeOs	0.39	1.17	-0.17
FeIr	0.87	1.49	0.58
FePt	1.14	1.76	1.16
FeAu	1.13	1.41	0.82

Table R18. $\Delta G_{*OH \rightarrow *OH-*OH}$ of the second $*OH$ adsorbed at different sites (M-M site, M-C site and M-M(anti) site) on the CoM@NC DACs.

CoM	$\Delta G_{*OH-*OH} - \Delta G_{*OH}$		
	M-M site	M-C site	M-M(anti) site
CoTi	-0.38	1.09	-0.79
CoV	-0.18	1.26	0.80
CoCr	0.23	2.13	0.62
CoMn	0.19	1.26	0.78
CoCo	1.05	3.37	3.22
CoNi	0.88	1.83	0.91
CoCu	1.41	1.49	0.37
CoZr	-0.47	1.62	0.86
CoNb	-0.22	1.10	1.02

CoMo	-0.02	1.38	-0.13
CoRu	0.27	1.80	-0.03
CoRh	1.12	1.45	0.56
CoPd	1.16	1.36	1.41
CoAg	0.72	1.65	0.33
CoHf	-0.65	1.27	0.47
CoTa	-0.32	1.09	0.68
CoW	0.07	1.24	-0.14
CoRe	0.01	1.33	-0.25
CoOs	0.22	1.55	0.13
CoIr	1.11	2.52	0.68
CoPt	1.30	1.93	1.61
CoAu	1.88	1.57	0.63

Table R19. $\Delta G_{*OH \rightarrow *OH \cdot *OH}$ of the second $*OH$ adsorbed at different sites (M-M site, M-C site and M-M(anti) site) on the NiM@NC DACs.

NiM	$\Delta G_{*OH \cdot *OH} - \Delta G_{*OH}$		
	M-M site	M-C site	M-M(anti) site
NiTi	0.03	1.28	0.01
NiV	0.17	1.35	0.10
NiCr	0.23	1.34	-0.03
NiMn	0.48	1.34	0.28
NiNi	1.28	1.18	1.19
NiCu	0.92	1.50	1.22
NiZr	-0.54	1.40	-0.08
NiNb	-0.74	1.16	0.00
NiMo	-0.04	1.41	0.22
NiRu	0.29	1.61	0.24
NiRh	1.04	1.56	0.43
NiPd	0.88	1.27	0.88
NiAg	0.80	1.87	0.82
NiHf	-0.59	1.21	-0.56
NiTa	-0.92	1.50	-0.22
NiW	-0.27	1.36	-0.11
NiRe	-0.19	1.37	0.05

NiOs	-0.06	1.53	0.11
NiIr	0.85	1.46	0.33
NiPt	1.11	1.38	1.13
NiAu	1.00	1.51	1.34

Table R20. $\Delta G^{*OH \rightarrow *OH} - \Delta G^{*OH}$ of the second *OH adsorbed at different sites (M-M site, M-C site and M-M(anti) site) on the CuM@NC DACs.

CuM	$\Delta G^{*OH \rightarrow *OH} - \Delta G^{*OH}$		
	M-M site	M-C site	M-M(anti) site
CuTi	-0.93	0.70	-0.78
CuV	0.05	1.46	0.11
CuCr	0.33	1.49	0.10
CuMn	0.38	1.28	0.33
CuCu	1.48	1.66	1.29
CuZr	-0.53	1.70	-0.04
CuNb	-0.63	1.35	-0.20
CuMo	-0.63	1.24	-0.28
CuRu	0.25	1.54	-0.22
CuRh	1.05	1.72	0.11
CuPd	1.10	1.51	1.20
CuAg	1.28	1.95	1.70
CuHf	-0.72	1.46	-0.48
CuTa	-0.92	1.26	-0.74
CuW	-0.88	1.22	-0.49
CuRe	-0.49	1.42	-0.32
CuOs	-0.02	1.44	-0.32
CuIr	0.73	1.62	-0.05
CuPt	1.21	1.37	1.19
CuAu	1.53	1.77	1.34

Revisions:

Page 11, Line 27:

“The above analysis on activity is based on the hypothesis that both *OH are adsorbed on the metal dimer, and whether alternative adsorption behavior is possible? For example, the second *OH adsorbs on the adjacent C site, or on the reverse side of the catalyst.^{48,55} As shown in

Supplementary Fig. 5a and Supplementary Tables 52-55, virtually the entire DACs, the second *OH prefers to adsorb at the M-M site compared to the M-C site, which circumvents the effect of the adjacent C site for *OH adsorption. Likewise, we also investigated the possibility of the second *OH adsorption on the other side of the DACs, which can be achieved for nearly half of the 86 DACs (Supplementary Fig. 5b and Supplementary Tables 52-55). However, although the above calculations indicate that the possibility of *OH adsorption on the reverse side, we used a monolayer model for DACs in this work, whereas experimentally synthesized graphene-based catalysts usually consist of multilayers,⁷⁸ and not only intercalation effects need to be considered, such structures also prevent *O-*O coupling.⁵⁵ On the other hand, different combinations of metal dimer have various O affinities without assurance that the *OH on the other side will not undergo further evolution (e.g., *O and *OOH), which complicates the picture and is not considered in this work.”

Supplementary Figure and Table:

Figure R14 has been included in Supplementary Fig. 5

Tables R17-R20 have been included in Supplementary Tables 52-55.

References, Page 28:

78. Luo, G., Jing, Y., Li, Y., Rational design of dual-metal-site catalysts for electroreduction of carbon dioxide. *J. Mater. Chem. A* **8**, 15809-15815 (2020).

Comment 3: The reason for proposing LOM or even OCM is the fact that *O-*O structures are generally more stable than OO* structures. Firstly, many of the selected DACs do not have this property, and secondly, the energy barrier of the step *O-*O→O₂ is generally higher, therefore, the author needs to calculate the transition state, or even the kinetic pathways of the OCM.

Response:

We thank the reviewer for the comment. Actually, the reason we proposed OCM in the manuscript is that the two adjacent active sites of DACs could provide the feasibility of *O-*O coupling. More specifically, the occurrence of OCM, not AEM, lies in the feasible adsorption of second *OH on the metal dimer, which avoided the generation of *OOH, rather than depends on the adsorption

configuration of *O₂. Regarding the reviewer's concern that many selected DACs are *OO structures more stable than *O-*O, the combination of O₂ with DACs can be classified into two configurations, the end-on configuration with superoxide character and the side-on configuration with peroxide character (*J. Catal.* **2023**, 417, 351–359), with all *O₂ configurations necessarily to be thermodynamically most stable, which excludes artificiality in the selection of configuration. The adsorption of O₂ in the present study involves both of the above conformations, as updated in Fig. 3 in the revised manuscript.

*O-*O coupling played very crucial role in OCM and we supplemented the calculation of *O-*O coupling barriers. Taking the 8 DACs with the highest activity as examples, the *O-*O coupling barriers in order as CoCu (0.04 eV) < CuPt (0.05 eV) < CuCu (0.06 eV) < CuPd (0.12 eV) = NiCu (0.12 eV) < NiPd (0.16 eV) < NiPd (0.28 eV) < CoPd (0.43 eV), which is in the same magnitude as the kinetic barriers shown in Figures R16 and R17 (*J. Mater. Chem. A*, **2022**, 10, 8309-8323; *J. Am. Chem. Soc.* **2017**, 139, 17281–17284). Note that Figure R17 demonstrates the process of *O₂ dissociation, and hence the inverse process (*O-*O coupling) is deduced. Referring to the O₂ dissociation (inverse process: *O-*O coupling) on DACs of the identical structure (*ACS Catal.* **2022**, 12, 3420–3429), our calculated configuration of the transition state as well as the *O-*O distance are similar to Figure R18 (*ACS Catal.* **2022**, 12, 3420–3429). It is believed that coupling barriers of this magnitude is kinetically favorable. Therefore, the above 8 DACs can be considered as promising catalysts for the fulfilment of *O-*O coupling along OCM.

Figure R15. Calculated kinetic barriers for the O-O coupling via OCM pathway on (a) CoCu@NC, (b) CoPd@NC, (c) NiCu@NC, (d) NiPd@NC, (e) NiPt@NC, (f) CuCu@NC, (g) CuPd@NC and (h) CuPt@NC. IS, TS, and FS denote the initial state, transition state and final state, respectively.

[Redacted]

Figure R16. Calculated kinetic barriers for the $*O+*O \rightarrow *O_2$ reaction via (a) Fe-Co path on FeCoN₆-gra, (b) Fe-C1 path on FeFeN₆-gra (c) Co-Co path on CoCoN₆-gra. IS, IS, TS, and FS denote the initial state, transition state and final state, respectively. (*J. Mater. Chem. A*, **2022**,10, 8309-8323)

Figure R17. Calculated kinetic barriers of O_2 dissociation pathway of ORR on (a) Fe SAs/N-C and (b) Co SAs/N-C. [Reprint (adapted) with permission from Jing Wang, Zhengqing Huang, Wei Liu, et al., Design of N-Coordinated Dual-Metal Sites: A Stable and Active Pt-Free Catalyst for Acidic Oxygen Reduction Reaction, *J. Am. Chem. Soc.* 139, 17281–17284 (2017). Copyright © 2017 American Chemical Society.]

Figure R18. Optimized structures, binding energies (E_b) and relative energies (E_{re}) of the O₂ activation on different M₁N₃-M₂N₃@NGr structured DACs. IS: initial state; TS: transition state; FS: final state. The O atoms in adsorbates are all represented by red spheres. [Reprint (adapted) with permission from Chuanyi Jia, Qian Wang, Jing Yang, et al., Toward Rational Design of Dual-Metal-Site Catalysts: Catalytic Descriptor Exploration, *ACS Catal.* 12, 3420–3429 (2022). Copyright © 2022 American Chemical Society.]

Revisions:

Page 10, Line 1:

Fig.3 has been revised below.

Fig. 3 Gibbs free energy diagrams and optimized configurations. **a-h** Gibbs free energy diagrams of NiPd@NC, CuPd@NC, CuPt@NC, CoPd@NC, NiPt@NC, NiCu@NC and the respective corresponding SACs for OER. **i** Optimized configurations of oxygenated intermediates represented as NiPd@NC.

Page 17, Line 4:

“Additionally, the kinetic process of *O-*O coupling is also critical for triggering OCM, and higher kinetic barriers reduce the potential for OCM to occur. Therefore, we further calculated the possible coupling processes based on the 8 heteronuclear DACs screened (Supplementary Fig. 17 and Table 80). The results show that all 8 DACs possessed low coupling barriers and by the same magnitude as reported in the literature.⁴⁸ As a result, *O-*O coupling cannot hinder OCM from

proceeding.”

Page 20, Line 17:

“The climbing image nudged elastic band (CI-NEB) method⁹⁴ combined with improved dimer method (IDM)⁹⁵ was employed to searched for transition states (TS).”

References, Pages 29 and 30:

94. Henkelman, G., Uberuaga, B. P., Jónsson, H., A climbing image nudged elastic band method for finding saddle points and minimum energy paths. *J. Chem. Phys.* **113**, 9901-9904 (2000).

95. Henkelman, G., Jónsson, H., A dimer method for finding saddle points on high dimensional potential surfaces using only first derivatives. *J. Chem. Phys.* **111**, 7010-7022 (1999).

Supplementary Figure and Table:

Figure R15 has been included in Supplementary Fig. 17.

Comment 4: The mechanism, OCM, that the author summarized, applies to both acidic and alkaline electrolytes. However, like the AEM, the potential-determining step is generally different for acidic and alkaline electrolytes, and the author needs to add discussions based on the calculation results.

Response:

We thank the reviewer for the comment. We have listed the generally accepted elementary steps involving the four electron transfer steps under different pH conditions for both AEM and OCM in the manuscript. To date, to the best of our knowledge, experimentally the effect of pH is mainly reflected in the stability of the catalyst and its resulting reconfiguration of the active site, rather than directly affecting the potential-determining step, e.g., most transition metal-based oxide catalysts can only survive in alkaline media and undergo oxidation and dissolution under acidic conditions (*Chem. Soc. Rev.*, **2015**,44, 2060-2086; *J. Phys. Chem. Lett.*, **2014**, 5, 2474-2478; *J. Electroanal. Chem. Interfacial Electrochem.*, **1984**, 172, 211-219), as well as the fact that many Ni-, Co-, Fe- and Mn-based OER catalysts undergo a surface reconstruction process during anodic polarization to form highly active oxyhydroxide species, which have been confirmed as the real active phase for alkaline OER reactions

(*Acc. Chem. Res.*, **2018**, 51, 2968–2977; *ACS Energy Lett.*, **2018**, 3, 2956–2966; *J. Mater. Chem. A*, **2020**, 8, 10096–10129; *Chem. Soc. Rev.*, **2020**, 49, 2196–2214; *Small*, **2019**, 15, 1901980). Therefore, we assume that different pH conditions may not directly contribute to potential-determining step.

For the calculation of DFT level, it is currently divided into the computational hydrogen electrode (CHE) model based on the constant charge method (CCM), and the constant potential model (CPM) (*ACS Catal.* **2023**, 13, 3, 1717-1725; *J. Phys. Chem. Lett.* **2021**, 12, 12230-12234; *Adv. Funct. Mater.* **2022**, 32, 2207110; *Chem. Sci.*, **2022**, 13, 6366), and the latter can include the effect of pH. CCM considered that the contribution of pH to the Gibbs free energy can be included by considering a term of the type $k_bT \ln 10 \times \text{pH}$, except for the non-electrochemical step of O₂ desorption. The pH shifts from $n = 0$ to 14 and the Gibbs free energy diagram is theoretically shifted upwards by $n \times 0.059$ eV, which means that the calculated limiting potential remains unchanged. In other words, the activity trend does not change with pH (*J. Am. Chem. Soc.* **2021**, 143, 20431–20441). A recently reported literature indicates that (phen₂N₂)FeCl catalysts treated with CPM exhibit high OER performance with the theoretical overpotential (η) of 0.32 V at pH = 1 (*Adv. Funct. Mater.* **2022**, 32, 2207110). Although its η increases with the values of pH, the η is still lower than 0.44 V at pH = 13 (Figure R19), which is similar to the results calculated with the CCM. In addition, it is demonstrated that the potential-determining step does not change under different pH conditions. Therefore, CCM, which is commonly adopted in the theoretical OER studies, still has the ability to quantitatively characterize reactivity (*Nat. Commun.* **2023**, 14, 112), and CPM is our research interest in the future.

[Redacted]

Figure R19. Free energy changes for OER of the (phen₂N₂)FeCl monolayer at (a) pH = 1 and (b) pH = 13 as a function of potential. (*Adv. Funct. Mater.* **2022**, 32, 2207110)

Revisions:

Page 21, Line 26:

“Note that, although the effect of pH on reactivity cannot explicitly taken into account with the computational hydrogen electrode model, according to a recent literature⁹⁸, we infer that the PDS was not affected by changing pH.⁹⁸”

References, Page 30:

98. Liu, T., Wang, Y., Li, Y., Two-dimensional organometallic frameworks with pyridinic single-metal-atom sites for bifunctional ORR/OER. *Adv. Funct. Mater.* **32**, 2207110 (2022).

Comment 5: It is necessary to carefully verify the rightness of the data for the OCM of NiPd@NC, because, the data do not theoretically match the given structures and mechanism. If the data are verified to be right, the author needs to analyze the electronic properties such as Bader charge for each step of the OCM of NiPd@NC.⁹⁸

Response:

We thank the reviewer for carefully reviewed the manuscript and pointed out our oversight. We apologize for the confusion between NiPd and NiPt. We have corrected this error in the revised version and listed the electronic structure information for NiPd, NiPt, CuPd and CuPt in Supplementary Information (Figures R20 and R21).

Figure R20. **a** The partial density of states (PDOS) for the Ni/Pd-d orbitals of Ni₂@NC, Ni@NC, NiPd@NC, Pd@NC and Pd₂@NC, respectively. **b** Crystal orbital Hamilton population (-COHP) for Ni₂@NC, Ni@NC, NiPd@NC, Pd@NC and Pd₂@NC, respectively. **c** Charge density differences ($\Delta\rho = \rho_{(\text{slab+ads})} - \rho_{(\text{slab})} - \rho_{(\text{ads})}$) of Ni₂@NC, Ni@NC, NiPd@NC, Pd@NC and Pd₂@NC, respectively, and

isosurface level = 0.004 e/Bohr³. Yellow: charge accumulation; cyan: charge depletion.

Figure R21. a-c Partial density of states (PDOS) of NiPt@NC, CuPd@NC, CuPt@NC and the corresponding homonuclear DACs as well as SACs, respectively. d-f Crystal orbital Hamiltonian groups (-COHP) of NiPt@NC, CuPd@NC, CuPt@NC and the corresponding homonuclear DACs and SACs, respectively.

Revisions:

Page 18, Line 22:

“The electronic structures of NiPt, CuPd and CuPt also have similar phenomena and regularities, as shown in Supplementary Fig. 21.”

Supplementary Figure and Table:

Figure R20 has been included in Supplementary Fig.18.

Figure R21 has been included in Supplementary Fig.21.

REVIEWER COMMENTS

Reviewer #1 (Remarks to the Author):

The authors have clarified some aspects raised by the referee and better discussed the novelty of their work. They have also performed several additional calculations to corroborate their findings. Therefore, the effort is largely appreciated. However, I must say that there are parts that are still unclear and somewhat contradictory, and given the high-level nature of the journal I still can not provide a positive recommendation. If the authors could address the following points the article could be suitable for Nat. Comm..

Based on the authors claims, they are proposing that i) the O-O coupling mechanism is preferred on Dual Atom Catalsts, ii) the formation of OH^*OH^* leads to scaling relationships different from those of the standard O^* and OOH^* OER intermediates, iii) a solid recipe for the catalyst optimization is proposed.

As I mentioned in the previous round, it was already reported that the O-O coupling mechanism is preferred [10.1016/j.jcat.2022.12.014] (on Single Atom Catalysts) and that OH^*OH^* brakes the scaling relationships [10.1021/acscatal.0c00815]. In this respect, the work provides a further step in this direction but the fact that previous works have already shown similar result is somewhat not sufficiently clear in the revised version of the work.

An important aspect to underline is that the nature of journal implies a clear-cut novelty and insight, and one could expect to find a clear recipe for the catalyst optimization, also because of the previous discussion. Instead, based on very large number of calculations one can extract some candidates but no descriptors (see point below) for future predictions are present.

Figure 6a shows that many cases have O^* more stable than OH^*OH^* , and this partly contradicts the first (i) conclusion, and most importantly it does not allows to use the free energy of OH^*OH^* as a descriptor. A clear explanation for this discrepancy is needed.

Reviewer #2 (Remarks to the Author):

The author answered all questions correctly and I suggest that this paper can be accepted.

Responses for the Nature Communications

Manuscript ID: NCOMMS-22-48108A

Title: Synergy of Dual-atom Catalysts Deviated from the Scaling Relationship for the Oxygen Evolution Reaction

At the outset, we would like to thank the Editor for providing us with one more opportunity to revise the manuscript. We are also very grateful to the reviewers for their time as well as valuable comments and suggestions. Provided below is our detailed point-to-point response to each question. The explanations to the comments from **Reviewer #1** are shown **in blue color**. The modifications provided in the revised manuscript and Supplementary Information are highlighted **with yellow background**.

Responses to the comments of reviewers:

Reviewer #1 (Remarks to the Author):

The authors have clarified some aspects raised by the referee and better discussed the novelty of their work. They have also performed several additional calculations to corroborate their findings. Therefore, the effort is largely appreciated. However, I must say that there are parts that are still unclear and somewhat contradictory, and given the high-level nature of the journal I still can not provide a positive recommendation. If the authors could address the following points the article could be suitable for Nat. Comm.

Response:

We are glad that the reviewer finds improvements in the revised manuscript, many of which are spurred from the useful comments and suggestions from the reviewer. We would also like to thank the reviewer for the review and comments on our manuscript again. All remaining concerns have been taken into careful consideration in this revision, and a point-to-point response to the specific comments could be found in the coming paragraphs below.

Comment 1: Based on the authors claims, they are proposing that i) the O-O coupling mechanism is preferred on Dual Atom Catalsts, ii) the formation of OH^*OH^* leads to scaling relationships different

from those of the standard O^* and OOH^* OER intermediates, iii) a solid recipe for the catalyst optimization is proposed.

As I mentioned in the previous round, it was already reported that the O-O coupling mechanism is preferred [10.1016/j.jcat.2022.12.014] (on Single Atom Catalysts) and that OH^*OH^* brakes the scaling relationships [10.1021/acscatal.0c00815]. In this respect, the work provides a further step in this direction but the fact that previous works have already shown similar result is somewhat not sufficiently clear in the revised version of the work.

Response:

Many thanks for the reviewer's comment. In terms of the two previously reported works, we completely agree with the reviewer on the significant finding of $*O-*O$ coupling mechanism and the breaking of the scaling relationship. Previous work [10.1016/j.jcat.2022.12.014] provided an in-depth and comprehensive assessment of all possible intermediates of SACs composed of ten metals on three different supports during OER and reliably derived the preferred reaction mechanism. Another literature [10.1021/acscatal.0c00815] mentioned that a new scaling relationship was identified for the unconventional $2OH^*$ mechanism on SACs in the ORR process which cannot be ignored in the future. In order to further carry forward the above highly valuable investigations, unconventional mechanisms (OCM and hybrid mechanisms) deserve more attention, i.e., **besides SACs, how it works in multi-active site systems?** To this end, we employed DACs as a prototype to exploit the advantages of OCM and even hybrid mechanisms in OER, demonstrating a unique scaling relationship derived from synergistic effects of dual-metal active sites. **This also clarifies the above misunderstanding comment from the reviewer that “i) the $*O-*O$ coupling mechanism is preferred on Dual Atom Catalsts”. Rather, our intention is to present how the feasibility of OCM on DACs is, and what conditions could possibly trigger the occurrence of OCM?**

As suggested by the reviewer, the significance of the previous works have been clearly acknowledged in the Introduction in the latest version of the revised manuscript. Furthermore, with regard to the comments referred to i), which is not mentioned in the manuscript, it is not our point of view (see below for a relevant discussion).

Revisions:

Page 4, Line 11:

“Recent studies have conclusively demonstrated that the contribution of unconventional mechanisms for electrocatalysis on the SACs cannot be overlooked, and that deviations in scaling relationships significantly affect electrochemical activity, for instance, the formation of dihydride in HER,⁵³ two *OH species in ORR and OER,^{54, 55} as well as superoxo and peroxy complexes in OER.⁵⁶ Furthermore, the *O-*O coupling mechanism (OCM), as one of the unconventional mechanisms which was proposed on SACs,⁵⁵ has also been shown with remarkably different activity at the dual-metal active site from that of the conventional adsorption evolution mechanism (AEM).^{48, 57} This is analogous to the direct coupling of adjacent oxygenated intermediates in the lattice oxygen mechanism (LOM), bypassing the generation of *OOH.^{58, 59}”

Page 4, Line 29:

“by which we could elucidate the fundamental factors in favor of OCM, popularize the advantages of OCM developed from valuable previous studies,^{54, 55} and further predict the promising candidates with much improved OER activity.”

Comment 2: An important aspect to underline is that the nature of journal implies a clear-cut novelty and insight, and one could expect to find a clear recipe for the catalyst optimization, also because of the previous discussion. Instead, based on very large number of calculations one can extract some candidates but no descriptors (see point below) for future predictions are present.

Response:

Many thanks to the reviewer for giving us the opportunity to clarify the strengths of our research. To the best of our knowledge, a large-scale investigation on dual-metal active sites for OER, particularly for the unconventional OCM mechanism, remains elusive. Therefore, our work contains four key findings:

- 1) Based on the large-scale screening of 86 DACs, a full picture of OER along the OCM is illustrated by means of an activity map, which facilitates the establishment of chemical intuitions to circumvent potentially poorly active DACs. Specifically, it is demonstrated that,

- even with different support, such as defective graphene, DACs incorporating early transition metals may not be suitable for achieving high OER activity through the OCM mechanism.
- 2) As distinct from SACs, the deviation of the scaling relationship caused by synergistic effects makes DACs that follow the OCM (or hybrid mechanism) superior candidates.
 - 3) A universal descriptor, $\Delta G_{*OH-*OH}$, was obtained by rational design, which qualitatively or even partially quantitatively evaluated the OER overpotential, and successfully predicted high performance of homonuclear DACs. Significantly, the $\Delta G_{*OH-*OH}$ descriptor as proposed would allow future researchers to quickly identify the activities of OER reactions along OCM by straightforward calculation of $\Delta G_{*OH-*OH}$. The applicability of the descriptors to $*O$ -selective DACs is also possible as will be described in detail in the following section.
 - 4) The trigger condition for OCM is a thermodynamic preference for the adsorption of a second $*OH$ rather than the adsorption of $*O$. In addition, the activity as well as the regularity of DACs with sufficient amounts along the hybrid mechanism have been studied systematically for the first time.

An activity descriptor of $\Delta G_{*OH}-\Delta G_{*O}$ for OER process via AEM mechanism has been previously reported, which in essence expects a constant rise of 1.23 eV for each elementary step resulting in an ideal $\eta = 0$. Similarly, the $\Delta G_{*OH-*OH}$ descriptor is rationally designed for OCM mechanism based on the screening of high-performance DACs as a reference. The optimal range of $\Delta G_{*OH-*OH}$ from 1.56 to 2.46 eV is also pertinent to a constant rise of 1.23 eV for each elementary step along the OCM and thus evaluating OER activity (as mentioned above). In addition, the applicability of the $\Delta G_{*OH-*OH}$ descriptor for the hybrid mechanism will be discussed in the next section.

Furthermore, in order to make our research more instructive for the actual OER process, as well as to further enhance the novelty and insight of our work, preliminary calculations with the constant potential method (CPM) have been carried out and complemented in this revised manuscript. To obtain the potential as well as pH-dependent energies, we performed the CANDLE (*J. Chem. Phys.* **2015**, 142, 064107) implicit solvation model in JDFTx code (*SoftwareX* **2017**, 6, 278–284). As an emerging cutting-edge technique, this has been used successfully in recent electrochemical studies (*Nat. Nanotechnol.* **2023**, 18, 160–167; *Nat. Catal.* **2022**, 5, 564–570; *Nat. Energy* **2019**, 4, 512–518; *Nat. Catal.* **2020**, 3, 98–106), especially the OER process including with Co-TiO₂ (*Nat. Catal.* **2021**, 4, 36–45), IrO₂ (*Nat. Commun.* **2021**, 12, 6007; *J. Am. Chem. Soc.* **2017**, 139, 149–155), Fe-doped γ -NiOOH

(*Proc. Natl Acad. Sci. USA* **2018**, 115, 5872–5877) and other metals doped into NiOOH (*J. Am. Chem. Soc.* **2018**, 140, 6745–6748). By using this method, our main findings are as follows:

- 1) The activity trends of NiPd@NC, CuPd@NC and CuPt@NC obtained under the CPM are identical to those of the constant charge method (CCM) under alkaline conditions. In addition, the change in pH triggered a variation in PDS, resulting in $\eta_{\text{CuPd}} (0.20 \text{ V}) > \eta_{\text{CuPt}} (0.18 \text{ V})$ under acidic conditions. Notwithstanding, the results suggest the three DACs are promising catalysts via OCM mechanism under both acidic and alkaline conditions (Figure R1 a-f).
- 2) As the applied potential increases, the maximum free energy change (ΔG_{max}) of the PDS gradually decreases until the overpotential is obtained at $\Delta G_{\text{max}} = 0$ (Figure R2). The oxygenated intermediates exhibit different binding strengths at different active centers, which is the principal factor contributing to the distinct activities of the three DACs under the same pH conditions.
- 3) With the pH decreased from 13 to 1, the relative position of the lines representing each elementary step in Figure R1 a-f shifted, particularly for the most potential PDS of the $*\text{OH} \rightarrow *\text{O} \rightarrow \text{O}_2$, which caused changes in the overpotential and even the PDS (Figure R1 a and b). Figure R1 g-i illustrates more visually the variation in the free energy diagrams under different potentials as well as pH conditions.

Figure R1. Variation of Gibbs free energy with potential for (a) NiPd@NC, (b) CuPd@NC and (c) CuPt@NC at pH=1. Variation of Gibbs free energy with potential for (d) NiPd@NC, (e) CuPd@NC and (f) CuPt@NC at pH=13. Free energy diagrams of (g) NiPd@NC, (h) CuPd@NC and (i) CuPt@NC at different potentials with pH = 1 and 13, and the corresponding $\Delta G_{*OH-*O \rightarrow *O_2}$ are labeled.

Figure R2. Free energy diagrams for (a) NiPd@NC, (b) CuPd@NC and (c) CuPt@NC at different potentials at pH=1 and 13 respectively.

Revisions:

Abstract, Page 2:

“Meanwhile, the effect of pH and potential on OER activity was clarified by employing grand canonical DFT (GC-DFT).”

Page 19, Line 18:

“pH-Dependent and Potential-Dependent OER Activity

The above results based on conventional constant charge methods (CCM) demonstrate that the synergistic effect between the dual-metal active sites in DACs can enhance OER activity. To better simulate the electrochemical interface and to examine the effect of pH as well as potential on the reaction activity, we took advantage of the JDFTx code of the grand canonical DFT (GC-DFT) for the constant potential method (CPM) calculations (more details in the Supplementary Information),⁸⁵ which have been proved useful in recent electrochemical studies.^{86, 87} As an example, with the screened high-performance NiPd@NC, CuPd@NC and CuPt@NC, we adjusted the absolute electrode potential to change the pH and electrochemical interface potential (referenced to RHE) as shown in Fig. 7 and Supplementary Tables 81-83. Interestingly, the activity order evaluated by the two methods of CCM (Fig. 3) and CPM was consistent for the three DACs under alkaline condition. In addition, the change in pH triggered a change in PDS, rendering the higher η for CuPd than CuPt under acidic conditions. The order of η for these three DACs is 0.08 V (NiPd) < 0.18 V (CuPt) < 0.20 V (CuPd) in acidic conditions, and 0.10 V (NiPd) < 0.18 V (CuPd)

< 0.28 V (CuPt) in alkaline conditions, respectively, indicating they are all potentially promising catalysts via OCM under both acidic and alkaline conditions.”

Page 21, Line 1:

“As shown in Figs. 7a-f, the free energy changes (ΔG) of the elementary steps decreases as the applied potential increases until the overpotential is determined with the maximum $\Delta G = 0$ (Supplementary Fig. 23), and this is analogous to a recent study of (phen₂N₂)FeCl monolayer.⁸⁸ Furthermore, the PDS of the three DACs is *OH-*O \rightarrow *O₂ (for NiPd@NC and CuPd@NC at pH = 1, it is *OH-*OH \rightarrow *OH-*O). The difference in activity of the three DACs derived from the different sensitivity of their oxygenated intermediates to the electrical potential, with the *OH-*O species being more significant (Supplementary Fig. 23). The decreasing trend of ΔG (*OH-*O \rightarrow *O₂) can be prominently observed in Figs. 7g-i with the gradual application of electrical potential. In terms of pH, another critical factor affecting activity, the considerable difference of 0.1 V in the overpotential of CuPt@NC under acidic and alkaline conditions can be attributed to the shift in the relative location of *OH-*O \rightarrow *O₂ (as denoted by the yellow lines in Figs. 7c and f). The PDS of NiPd@NC and CuPd@NC altered from *OH-*O \rightarrow *O₂ to *OH-*OH \rightarrow *OH-*O as the alkaline to acidic conditions changed (Figs. 7a and b). Consequently, both pH and potential have significant contribution to the electrochemical process, with the influence on the PDS and overpotential. Nevertheless, the superior OER activity for the screened three DACs can be appraised.”

Page 20, Line 4:

Figure R1 has been included as Fig. 7 in the revised manuscript.

Discussion, Page 22:

“Further, CPM considering surface charge reveals pH-dependent and potential-dependent OER activity on diatomic catalysts, indicating the selected three DACs of NiPd@NC, CuPd@NC and CuPt@NC are promising candidates under both acidic and alkaline conditions.”

References, Page 29:

85. Sundararaman, R., Letchworth-Weaver, K., Schwarz, K. A., Gunceler, D., Ozhabes, Y., Arias, T. A., JDFTx: Software for joint density-functional theory. *SoftwareX* **6**, 278-284 (2017).
86. Liu, C., *et al.*, Oxygen evolution reaction over catalytic single-site Co in a well-defined brookite TiO₂ nanorod surface. *Nat. Catal.* **4**, 36-45 (2020).
87. Xie, Y., *et al.*, High carbon utilization in CO₂ reduction to multi-carbon products in acidic media. *Nat. Catal.* **5**, 564-570 (2022).
88. Liu, T., Wang, Y., Li, Y., Two-dimensional organometallic frameworks with pyridinic single-metal-atom sites for bifunctional orr/oer. *Adv. Funct. Mater.* **32**, 2207110 (2022).

Supplementary Information, Methods:

“The grand canonical ensemble DFT simulations were performed using the JDFTx software,¹⁵ and the electron exchange-correlation energy was evaluated within the GGA-PBE functional, consistent with the calculation in VASP. The GBRV pseudopotentials and a plane wave energy cut-off of 20 Hartree was employed. The linear PCM solvation model (CANDLE) to simulate the liquid environment, with 0.1 M K⁺ and F⁻ ions.¹⁶ The Brillouin zone was sampled using a 2×2×1 Monkhorst-Pack k-point mesh and a convergence threshold of 1×10⁻⁶ Hartree was set for the total electronic energy. The effective regulation of pH and potential can be described by the following relation:

$$U \text{ (V/RHE)} = U \text{ (V/SHE)} + 0.0592 \times \text{pH} \quad (28)$$

$$U \text{ (V/SHE)} = -4.66 \text{ V} - U_{\text{applied}} \text{ (V/SHE)} \quad (29)$$

where -4.66 V is the absolute electrode potential with SHE as reference,¹⁷ and the U_{applied} is the applied electrode potentials of the targets.”

Supplementary Figure:

Figure R2 has been included as Supplementary Fig. 23.

Supplementary Table:

Tables S81-83 are added below.

Table 81. Free energy changes for each intermediate of NiPd@NC under different conditions.

Intermediate	$\Delta G(\text{eV})$					
			U = 0 V vs.RHE		U = 1.23 V vs.RHE	
	Neutral	Candle	pH = 1	pH = 13	pH = 1	pH = 13
*OH	0.868	0.799	0.509	0.676	0.319	0.427
*OH-*OH	1.371	1.179	1.217	1.220	1.368	1.287
*OH-*O	2.799	2.403	2.288	2.042	2.665	2.471
*O ₂	3.975	3.715	3.714	3.624	3.927	3.815

Table 82. Free energy changes for each intermediate of CuPd@NC under different conditions.

Intermediate	$\Delta G(\text{eV})$					
			U = 0 V vs.RHE		U = 1.23 V vs.RHE	
	Neutral	Candle	pH = 1	pH = 13	pH = 1	pH = 13
*OH	1.074	0.756	0.505	0.564	0.092	0.396
*OH-*OH	2.003	1.420	1.342	1.196	1.270	1.399
*OH-*O	3.256	2.581	2.510	2.151	2.670	2.680
*O ₂	4.975	4.537	4.111	3.965	3.969	4.129

Table 83. Free energy changes for each intermediate of CuPt@NC under different conditions.

Intermediate	$\Delta G(\text{eV})$					
			U = 0 V vs.RHE		U = 1.23 V vs.RHE	
	Neutral	Candle	pH = 1	pH = 13	pH = 1	pH = 13
*OH	1.081	0.775	0.523	0.597	0.088	0.406
*OH-*OH	2.120	1.558	1.436	1.306	1.335	1.481
*OH-*O	3.198	2.539	2.399	2.066	2.522	2.551
*O ₂	4.908	4.486	4.089	3.943	3.962	4.112

Supplementary Information, References:

16. Sundararaman, R., Letchworth-Weaver, K., Schwarz, K. A., Gunceler, D., Ozhabes, Y., Arias, T. A., JDFTx: Software for joint density-functional theory. *SoftwareX* **6**, 278-284 (2017).
17. Sundararaman, R., Goddard, W. A., The charge-asymmetric nonlocally determined local-electric (CANDLE) solvation model. *J. Chem. Phys.* **142**, 064107 (2015).
18. Trasatti, S., Structure of the metal/electrolyte solution interface: new data for theory. *Electrochim. Acta* **36**, 1659-1667 (1991).

Comment 3: Figure 6a shows that many cases have O* more stable than OH*OH*, and this partly contradicts the first (i) conclusion, and most importantly it does not allow to use the free energy of OH*OH* as a descriptor. A clear explanation for this discrepancy is needed.

Response:

Thank you very much for this comment. Firstly, as clarified in Response to Comment 1, the conclusion (i) is actually not mentioned in our manuscript. Instead, our high-throughput calculations indicated that only 11 out of 105 DACs are highly active via OCM with the possibility of breaking the 0.37V bottleneck.

Second, we want to clarify the point that the cases with *O more stable in Fig. 6a are less than half (*OH-*OH: *O = 56:49). Specifically, it would be arbitrary to completely discount the possibility of AEM on DACs, and we prefer to remind relevant researchers that OCM or even hybrid mechanisms cannot be ignored, as mentioned in the literature [10.1016/j.jcat.2022.12.014], and note that the 10 highly active DACs screened prefer OCM (except for CoCu).

The reviewer's concern about the potential unsuitability of employing $\Delta G_{*OH-*OH}$ as a descriptor for characterizing the *O-selective DACs is legitimate. We will address this concern in two aspects:

- 1) **The volcano plot of Fig. 5a in the manuscript applies in general to the OER process along the OCM (strictly speaking, not other mechanism).** In the absence of knowledge of which mechanism, a certain DAC follows, the OCM can be preselected first. It is sufficient to obtain $\Delta G_{*OH-*OH}$ (1.56 eV - 2.46 eV) to judge the reactivity along the OCM. Following such an efficient screening, it is reasonable and necessary to evaluate the *O-selective DACs along the hybrid mechanism. With the narrowed scope of the screening, the second round of identifying *O-selective DACs is efficient and accurate. This can be reflected in the case of CoCu@NC,

which is prejudged as potential active catalyst along OCM, and the secondary selection indicates the hybrid mechanism via *O-selective path is preferred (although without changing overpotential).

- 2) **We further conclude that volcano plots employing $\Delta G^{*OH-*OH}$ as the descriptor can still be exploited to qualitatively ascertain the overpotential of DACs, notwithstanding for *O-selective DACs (Figure R3a).** Specifically, the blue dots, as well as the volcano curve derived from Fig. 5a in the manuscript, are DACs with *O selectivity. To more accurately characterize the OER activity of these DACs with *O selectivity, we repositioned these DACs with ΔG^{*O} as the descriptor (the purple dots). It was observed that DACs with ΔG^{*O} as descriptor remained on the original volcano curve, which was attributed to the fact that the $\Delta G^{*OH}/\Delta G^{*OH-*OH}$ scaling relationship virtually overlapped with that of $\Delta G^{*OH}/\Delta G^{*O}$ (Figure R3b).

In summary, on the one hand $\Delta G^{*OH-*OH}$ is generally applicable to OER processes along the OCM mechanism, and a secondary screening enables further rigorous identification of high-performance DACs with *O selectivity along hybrid mechanisms (e.g. CoCu). On the other hand, the fact that the $\Delta G^{*OH}/\Delta G^{*OH-*OH}$ scaling relationship and that of $\Delta G^{*OH}/\Delta G^{*O}$ virtually overlap rendered the DACs with ΔG^{*O} as descriptor still positioned on the original volcano curve. In other words, the volcano plot derived from $\Delta G^{*OH-*OH}$ as the descriptor still *roughly* determines the *O-selective DACs (e.g. CoAu) with regard to such accuracy, and the poorly active *O-selective DACs can be eliminated. **Therefore, in the cases of *O-selective DACs without the favorable formation of *OH-*OH intermediate, the activity descriptor of the volcano plot, following hybrid mechanisms, could be replaced with ΔG^{*O} ($1.56 \text{ eV} < \Delta G^{*O} < 2.46 \text{ eV}$). This is acceptable as a *rough preliminary* criterion for activity, and the more accurate scope of ΔG^{*O} needs to be explored in further detail.**

Figure R3. (a) Volcano plot of DACs with *O-selectivity (or following the hybrid mechanism). Blue represents $\Delta G_{*OH-*OH}$ as the descriptor and purple represents ΔG_{*O} as the descriptor. (b) Scaling relationships of $\Delta G_{*OH}^*/\Delta G_{*OH-*OH}^*$ (blue) and $\Delta G_{*OH}^*/\Delta G_{*O}^*$ (purple) for DACs with *O selectivity (or following the hybrid mechanism), respectively.

Revisions:

Page 17, Line 8:

“A critical question arises here: given that approximately half of the DACs exhibited *O selectivity, is it still valid to use $\Delta G_{*OH-*OH}^*$ in Fig. 5a as a descriptor to assess their reactivity? On the one hand $\Delta G_{*OH-*OH}^*$ is generally applicable to OER processes along the OCM, and a secondary screening enables further rigorous identification of high-performance DACs with *O selectivity along hybrid mechanisms (e.g. CoCu). On the other hand, the fact that the $\Delta G_{*OH}^*/\Delta G_{*OH-*OH}^*$ scaling relationship and that of $\Delta G_{*OH}^*/\Delta G_{*O}^*$ virtually overlap rendered the DACs with ΔG_{*O}^* as descriptor still positioned on the original volcano curve (Supplementary Fig. 18). In other words, the volcano plot derived from $\Delta G_{*OH-*OH}^*$ with the screening scope from 1.56 eV to 2.46 eV still roughly determines the *O-selective DACs (e.g. CoAu) with regard to such accuracy, and the poorly active *O-selective DACs can be eliminated.”

Supplementary Figure:

Figure R3 has been included in Supplementary Fig. 18.

Although this could be the last chance provided by the Editor, from the above justifications and revisions, we have strong confidence that the novelty and significance of the present study meets the requirement of *Nat. Commun.* We hope we could convince that the mechanistic insights reported in this manuscript will be appealing to the generalized readers of *Nat. Commun.* We really appreciate the reviewer's constructive comments/suggestions to further sharpening our mind in this interesting research topic.

Many thanks.

REVIEWERS' COMMENTS

Reviewer #1 (Remarks to the Author):

I have read with interest the revised version of the manuscript.
After careful examination I consider the manuscript acceptable for publication.
The authors claims, conclusions and novelty are balanced.

Responses for the Nature Communications

Manuscript ID: NCOMMS-22-48108B

Title: Synergy of Dual-atom Catalysts Deviated from the Scaling Relationship for Oxygen Evolution Reaction

Responses to the comments of reviewers:

Reviewer #1 (Remarks to the Author):

I have read with interest the revised version of the manuscript.

After careful examination I consider the manuscript acceptable for publication.

The authors claims, conclusions and novelty are balanced.

Response:

Thank you very much to Reviewer for reviewing our revised manuscript and providing a positive evaluation. The reviewer's comments and suggestions have helped us make our research more comprehensive, rigorous and convincing. We highly value the opinions of reviewers. The reviewer believes that our arguments, conclusions and innovations are quite balanced and reasonable. This shows that we have made the right modifications and supplements in the revision of the paper to improve the quality. Thank you again to the reviewer for the review and valuable opinions.